# Histone modifications regulate pioneer transcription factor cooperativity

Kalyan K. Sinha[1], Silvija Bilokapic[1], Yongming Du[1], Deepshikha Malik[1] & Mario Halic[1✉]

Pioneer transcription factors have the ability to access DNA in compacted chromatin[1]. Multiple transcription factors can bind together to a regulatory element in a cooperative way, and cooperation between the pioneer transcription factors OCT4 (also known as POU5F1) and SOX2 is important for pluripotency and reprogramming[2–4]. However, the molecular mechanisms by which pioneer transcription factors function and cooperate on chromatin remain unclear. Here we present cryo-electron microscopy structures of human OCT4 bound to a nucleosome containing human *LIN28B* or n*MATN1* DNA sequences, both of which bear multiple binding sites for OCT4. Our structural and biochemistry data reveal that binding of OCT4 induces changes to the nucleosome structure, repositions the nucleosomal DNA and facilitates cooperative binding of additional OCT4 and of SOX2 to their internal binding sites. The flexible activation domain of OCT4 contacts the N-terminal tail of histone H4, altering its conformation and thus promoting chromatin decompaction. Moreover, the DNA-binding domain of OCT4 engages with the N-terminal tail of histone H3, and post-translational modifications at H3K27 modulate DNA positioning and affect transcription factor cooperativity. Thus, our findings suggest that the epigenetic landscape could regulate OCT4 activity to ensure proper cell programming.

DNA-binding transcription factors (TFs) target distinct DNA sequences at gene regulatory regions, thus ensuring specificity in transcription machinery assembly[1,5]. DNA packaging into nucleosomes can hinder TF binding to target sequences[6], but a small set of so-called pioneer TFs can access DNA even within compacted chromatin[3,7–10]. Once bound to their target sites, pioneer TFs can facilitate the recruitment of other TFs by creating accessible chromatin, a property that underlies their function as master regulators in embryo development, cell differentiation and reprogramming. In fact, overexpression of four pioneer TFs – OCT4, SOX2, KLF4 and MYC – promotes the reprogramming of cells to pluripotency[2], with OCT4 expression being necessary and sufficient to reprogram cells[4,11]. In vitro SOX2, KLF4 and MYC bind to nucleosomes more efficiently in the presence of OCT4 (ref. 8), and cooperativity between OCT4 and SOX2 is critical for early development and reprogramming[12–18], but the molecular mechanisms involved remain unclear.

OCT4 has two DNA-binding domains: OCT4-POU$_S$ and OCT4-POU$_{HD}$. Previous X-ray structures showed the two domains wrapping around naked DNA, but such a binding mode would be incompatible with the nucleosome architecture[19,20]. In recent cryo-electron microscopy (cryo-EM) work[21], OCT4-POU$_S$ and the DNA-binding domain of SOX2 were seen unwrapping a nucleosome containing binding sites for those TFs inserted into the DNA positioning sequence 601 (ref. 22). The inserts were placed to promote optimal binding and stability of the complex, but the 601 sequence is known to suppress the nucleosome dynamics that are typical of biologically relevant sequences[22]. In recent efforts to capture OCT4 bound to a nucleosome with an endogenous DNA

sequence, the density for OCT4 could not be observed[23,24]. Hence, a structure of OCT4 in complex with a nucleosome containing a physiologically relevant DNA sequence remained elusive, limiting our mechanistic understanding of pioneer TF function. For instance, although OCT4 and other TFs bind to nucleosomes, it remains unclear whether they interact with histones and whether epigenetic marks would affect that interaction. Previous crosslinking and mass spectrometry analyses of the reconstituted OCT4–nucleosome complex with endogenous DNA have shown that OCT4 binds near to the N-terminal tail of histone H3 (ref. 24), which would require proper positioning of the DNA-binding site on the nucleosome. Moreover, chromatin occupancy by OCT4 correlates with histone marks found in enhancers, such as H3K27ac and H3K4me1, but silent marks such as H3K27me3 are also found at OCT4-binding sites[25–30]; it remains to be determined whether and how those modifications regulate OCT4 binding.

To address these gaps, we present cryo-EM structures of human OCT4 bound to nucleosomes containing DNA sequences from human *LIN28B* or near the matrilin 1 gene (n*MATN1*) loci, along with biochemistry assays. The *LIN28B* sequence has three binding sites for OCT4, as well as binding sites for SOX2, KLF4 and MYC[8], whereas n*MATN1* has multiple OCT4-binding sites. Both sequences are thus an ideal platform to study cooperative assembly of multiple pioneer TFs.

## OCT4 binding to the *LIN28B* nucleosome

To investigate the mechanism for cooperativity between the pioneer TFs OCT4 and SOX2, we assembled a complex containing full-length

[1]Department of Structural Biology, St. Jude Children's Research Hospital, Memphis, TN, USA. ✉e-mail: mario.halic@stjude.org

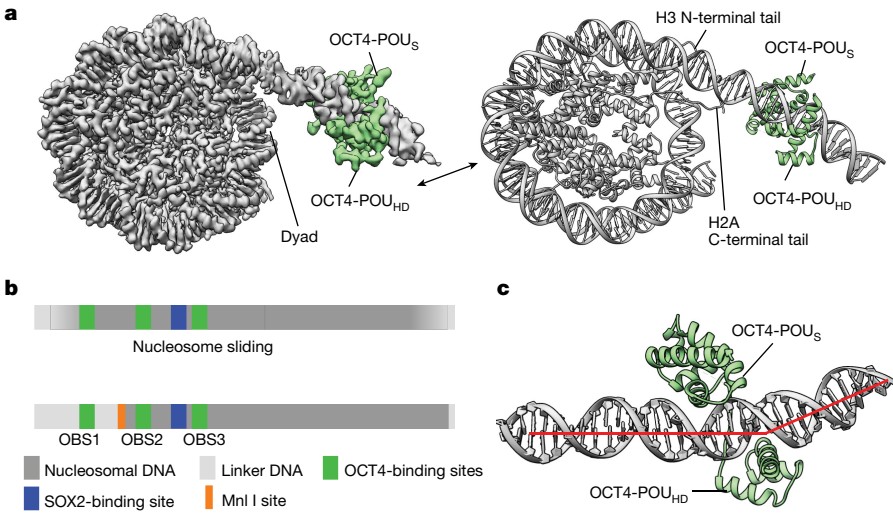

**Fig. 1 | OCT4 binds to the nucleosome at the exposed DNA site. a**, A composite cryo-EM map (left) and the structural model (right) of human OCT4 (green) bound to a nucleosome (grey) assembled with a 182-bp DNA fragment from the *LIN28B* locus. **b**, Schematic representation of DNA positioning on the *LIN28B* nucleosome. Binding sites for OCT4 (OBS1, OBS2 and OBS3) and SOX2, and the MnI I restriction site are shown. The nucleosome is 'fuzzy' as the DNA adopts multiple positions due to spontaneous sliding (top). OCT4 binding stabilizes DNA at a defined position on the nucleosome (bottom). **c**, Close-up view of OCT4 (green) bound to the nucleosomal DNA (grey; the red line shows the path of the DNA helix axis), showing the kink in the linker DNA introduced by OCT4-POU_HD.

human OCT4 and SOX2 and a nucleosome with DNA from the human *LIN28B* locus[31–33]. This DNA fragment contains three binding sites for OCT4 (OBS1–3) and one for SOX2 (ref. 8). Using native gel electrophoresis assays, we observed an association of OCT4 and SOX2 with nucleosomes assembled with *LIN28B* DNA fragments that were 162-bp or 182-bp long (Extended Data Fig. 1a). However, using cryo-EM analyses, we could only visualize the proteins bound to the 182-bp nucleosome, which indicates that the complex on the nucleosome with shorter DNA is less stable[24]. Hence, for the remainder of this work, we exclusively used the 182-bp nucleosome.

The initial cryo-EM reconstructions showed a density bound to the linker DNA (Extended Data Table 1 and Extended Data Fig. 1b–h), but the resolution was limited because of flexibility of the complex. To improve the resolution, we used focused classification followed by local search refinements and obtained maps with resolutions of 2.8 Å in the nucleosome portion (Extended Data Fig. 1g–i) and of 3.9 Å for a 30-kDa region of OCT4 bound to linker DNA (Extended Data Fig. 2a–g). We did not observe clear density for SOX2, which suggests that it might have dissociated during sample preparation. The two maps had sufficient overlapping densities to allow assembly of a composite map and model (Fig. 1a, Extended Data Table 1 and Extended Data Fig. 2h). In the structure, OCT4 is bound to the linker DNA near the nucleosome entry–exit site (Fig. 1a and Extended Data Fig. 2h), in agreement with previous crosslinking mapping of reconstituted complexes[24]. Of note, the DNA bases are well resolved along the nucleosome-wrapped region, indicating minimal movement of the *LIN28B* sequence in complex with OCT4 (Extended Data Fig. 1i). The high resolution of the nucleosomal DNA allowed us to precisely position the sequence, with OBS1 placed at the exact location of the OCT4 density (Fig. 1a and Extended Data Figs. 1i and 2i).

To determine whether OCT4 stabilizes DNA positioning on the *LIN28B* nucleosome, we determined a cryo-EM structure of that same nucleosome in the absence of any TFs (Extended Data Table 1). This structure had an overall resolution of 3.1 Å, similar to the OCT4-bound nucleosome, but the DNA bases were not well resolved and hence the DNA position could not be determined (Extended Data Fig. 3a–f). We also observed that in some particle classes, the linker DNA protrudes from both sides of the histone octamer (Extended Data Fig. 3g), whereas in the OCT4-bound structure, it protrudes from only one side. Together, these observations indicate that the *LIN28B* DNA could adopt several positions on the free nucleosome (Extended Data Fig. 3g), in contrast to its well-defined positioning in the OCT4-bound nucleosome (Fig. 1a). These findings are consistent with in vivo data showing that the nucleosome at the *LIN28B* locus is 'fuzzy' and occupies approximately 200 bp (ref. 34). Thus, the naturally occurring *LIN28B* sequence is able to move along the histone octamer, transiently exposing OBS1. Once OCT4 binds to OBS1, it traps the DNA in that position and stabilizes the otherwise flexible linker DNA into a more defined conformation (Fig. 1b and Extended Data Fig. 4a). Of note, we observed that hexasomes are threefold more abundant in OCT4-bound samples than in the free *LIN28B* nucleosomes (Extended Data Figs. 1g and 3f).

In the structure, both DNA-binding domains of OCT4 engage with the *LIN28B* nucleosome: OCT4-POU_S is bound to the linker DNA, close to the nucleosome dyad, whereas OCT4-POU_HD is located distally from the nucleosome (Fig. 1a and Extended Data Fig. 2h). These observations are consistent with the requirement for both OCT4 DNA-binding domains to efficiently bind to chromatin in vivo[21] (Fig. 1a,c). The OCT4 interactions with DNA in our structure are overall similar to those observed in the crystal structure of OCT4 bound to naked DNA[19], but we observed that OCT4-POU_HD introduces a kink in the linker DNA, due to arginine residues widening the DNA major groove (Fig. 1c and Extended Data Figs. 2g and 4b,c); such DNA distortion could disrupt local chromatin organization and affect binding of other proteins. Overall, the OCT4–DNA interactions in our structure differ considerably from those seen in the cryo-EM structure of OCT4 bound to a 601-based nucleosome, in which only OCT4-POU_S was observed to interact with the nucleosome[21].

## OCT4 modifies H4 tail conformation

Having shown that OCT4 binding stabilizes the positioning of nucleosomal DNA, we examined whether it induced other changes to the nucleosome structure. We observed in our cryo-EM data that the H4 N-terminal tail on the OCT4 proximal side of the nucleosome adopts multiple conformations, whereas the H4 tail on the opposite side is predominantly found in the canonical conformation, following the DNA path at SHL2 (refs. 35,36) (Extended Data Fig. 4d). Further image

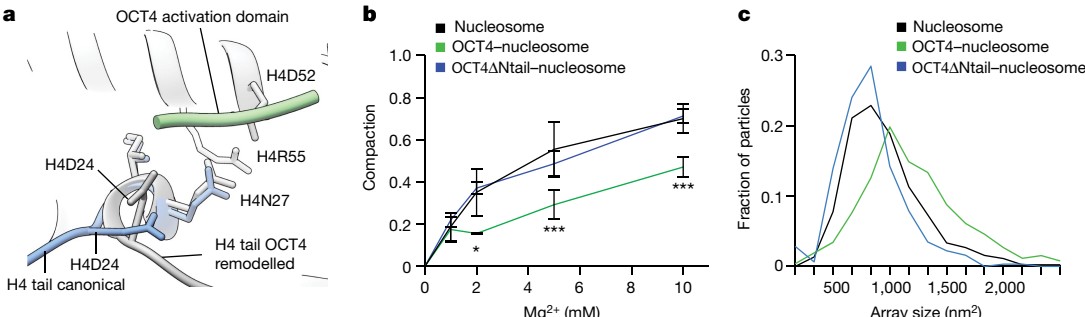

**Fig. 2 | OCT4 activation domain remodels the N-terminal tail of histone H4.** **a**, Overlay of the two distinct conformations of the H4 tail on the OCT4 proximal side of the nucleosome: canonical conformation (blue) and OCT4-remodelled conformation (grey). The interaction of the disordered region of OCT4 (green) with the H4 tail rearranges Asp24, leading to repositioning of the whole tail. **b**, Mononucleosome compaction by $Mg^{2+}$, assessed by native gel electrophoresis; a representative gel is shown in Extended Data Fig. 5a. Compaction was quantified by the reduction in the intensity of the nucleosome band, due to nucleosome precipitation. Data shown are mean ± s.e.m. of four independent measurements. *$P = 0.04$ for 2 mM $Mg^{2+}$, ***$P = 0.006$ for 5 mM $Mg^{2+}$ and ***$P = 1.7 \times 10^{-9}$ for 10 mM $Mg^{2+}$, one-sided Student's $t$-test comparing OCT4-bound nucleosome to nucleosome. **c**, Quantification of negative-stain EM data showing $Mg^{2+}$-induced compaction of nucleosome arrays assembled on a 1,022-bp-long DNA fragment from the *LIN28B* locus. The graph shows the distribution of the area occupied (size) by nucleosome arrays, measured from micrographs with the different samples; 300–450 arrays were analysed per sample. Representative micrographs are shown in Extended Data Fig. 5c.

classification revealed two major conformations for the H4 tail on the OCT4 proximal side of the nucleosome: the first one resembles the canonical conformation, whereas in the second one, the H4 tail is rotated 90° towards SHL1 and an additional density can be seen interacting with the H4 tail and α2 helix (Fig. 2a and Extended Data Fig. 4d–f). This density was not observed on the *LIN28B* nucleosome or on the OCT4 distal side of the OCT4-bound nucleosome, and we hypothesized that it could originate from the OCT4 activation domain, which consists of flexible N-terminal and C-terminal regions. The density alters the conformation of Asp24 at the beginning of the H4 tail, which in turn changes the conformation of the whole H4 tail (Fig. 2a), moving residues that are essential for chromatin compaction by more than 30 Å and potentially disrupting interactions between nucleosomes.

These observations suggest that OCT4 binding affects the interactions between two nucleosomes. To examine this effect further, we induced nucleosome compaction with $Mg^{2+}$ (refs. 37,38) and found that the presence of OCT4 substantially reduced association between mononucleosomes as assessed by native gel electrophoresis (Fig. 2b and Extended Data Fig. 5a). We also assembled chromatin arrays using a longer 1,022-bp DNA fragment from the *LIN28B* locus and examined their compaction with $Mg^{2+}$ by negative-stain EM imaging. We found that the nucleosome arrays are more open in the presence of OCT4 (Fig. 2c and Extended Data Fig. 5b,c).

Finally, we tested the roles of the N-terminal and C-terminal flexible regions of OCT4 on chromatin decompaction by deleting them individually. Neither deletion reduced the interaction of OCT4 with the *LIN28B* nucleosome (Extended Data Fig. 5d). However, OCT4 lacking the N-terminal region lost the ability to reduce nucleosome compaction and internucleosome interactions, whereas deletion of the C-terminal tail did not affect those properties (Fig. 2b,c and Extended Data Fig. 5e). Together, our structural and biochemical data suggest that the N-terminal region of OCT4 remodels the H4 tail and contributes to chromatin decompaction.

## H3K27ac increases OCT4 cooperativity

In our structure of the OCT4-bound nucleosome, both the OCT4-binding sites OBS2 and OBS3 have partial internal motifs exposed, which would allow binding of OCT4-POU$_{HD}$ to OBS2 and OCT4-POU$_S$ to OBS3 (Fig. 3a). Binding to partial DNA motifs has been previously proposed[8] and our data suggest that DNA positioning induced by binding of OCT4 to OBS1 facilitates binding of additional OCT4 molecules to their internal sites.

To test this hypothesis, we mutated each of the OCT4-binding sites in the *LIN28B* sequence and examined binding of OCT4 to nucleosomes by native gel electrophoresis. This setup allows us to distinguish nucleosomes with one or more OCT4 bound (Extended Data Fig. 5f). With the wild-type *LIN28B* nucleosome, we detected strong binding of one OCT4 but also a second and weaker binding of the third OCT4 (Extended Data Fig. 5f,g). Mutation of either OBS2 or OBS3 did not affect the formation of a complex with one OCT4 bound, indicating that the first OCT4 binds to OBS1 as in the wild-type *LIN28B* nucleosome, whereas a second OCT4 binds to either OBS2 or OBS3 (Extended Data Fig. 5h,i). By contrast, when we mutated OBS1 to generate the *LIN28B*-1M nucleosome, OCT4 binding to OBS2/3 was considerably reduced compared with the wild-type *LIN28B* nucleosome (Fig. 3b). Thus, OCT4 binding to OBS2/3 is stimulated when a first OCT4 is bound to OBS1, which is in agreement with our structural data showing that binding of OCT4 to OBS1 stabilizes nucleosomal DNA and exposes partial motifs in internal OBS2/3 sites.

We turned our attention back to OCT4 DNA-binding domains bound to OBS1 and observed interactions between OCT4-POU$_S$ and histones H3 and H2A. The tip of helix 1 (residues 159–163) contacts the C-terminal tail of histone H2A and the N-terminal tail of histone H3; the latter is also contacted by small helix 5 (residues 213–222) (Fig. 3c and Extended Data Fig. 5j). The dipole moment of helix 1 and negatively charged helix 5 together form an acidic patch on OCT4 that faces the nucleosomal dyad and interacts with positively charged histone tails there, mediating additional interaction between the OCT4 DNA-binding domain and the nucleosome (Fig. 3d).

In our structure of the OCT4-bound nucleosome, histone H3K27 is in close proximity to the acidic patch of OCT4, specifically to the tip of helix 1, suggesting a potential electrostatic interaction between positively charged H3K27 and the negative dipole moment of helix 1 (Fig. 3c,d). This observation prompted us to examine whether H3K27 modifications would affect OCT4 binding to the nucleosome. H3K27ac is an active mark associated with enhancers that was found to colocalize with OCT4 on chromatin[26-28]. Acetylation of H3K27 would neutralize the positive charge of the Lys residue and would be expected to affect the interaction between the histone H3 tail and the OCT4 acidic patch. To test this possibility, we assembled nucleosomes with H3K27ac-modified histone H3 and examined binding of OCT4 by native gel electrophoresis (Fig. 3e and Extended Data Fig. 6a). The H3K27ac modification showed only a small effect on binding of the first OCT4, which directly interacts with the H3K27 residue, indicating that this interaction is not required for stability of the OCT4–nucleosome complex (Fig. 3e). However,

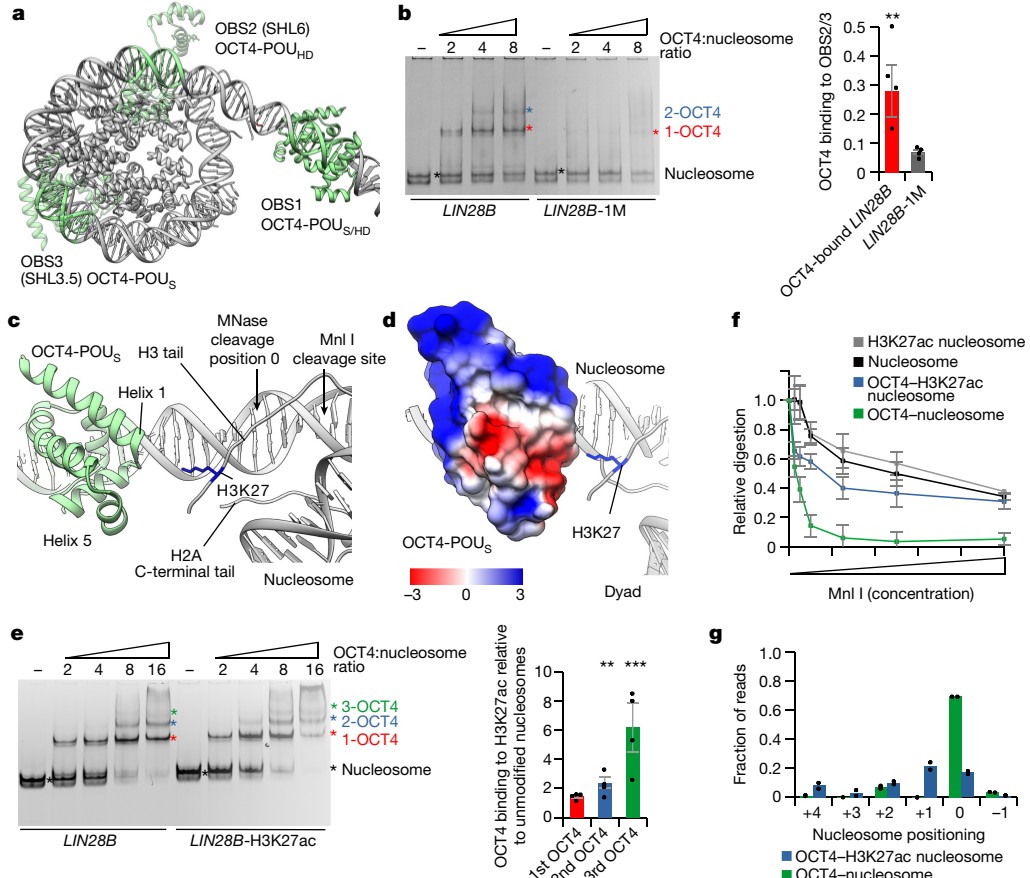

**Fig. 3 | Histone modifications modulate OCT4 cooperativity. a**, Model of OCT4 bound to the *LIN28B* nucleosome. OCT4-binding sites are in green, OCT4 bound to OBS1 is in solid green, and the OCT4 structure superimposed on OBS2 and OBS3 is in transparent green. **b**, Representative native gel electrophoresis showing OCT4 binding to *LIN28B* or *LIN28B*-1M nucleosomes (left). The asterisks mark the number of OCT4 bound: the nucleosome is in black, 1-OCT4 (one OCT4 molecule bound) is in red, and 2-OCT4 (two OCT4 molecules bound) is in blue. Band composition was validated by immunoblotting (Extended Data Fig. 5g). Quantification of binding of OCT4 to OBS2 and OBS3 is also shown (right). Data shown are mean ± s.e.m., *n* = 4 independent experiments; **P = 0.008, one-sided Student's *t*-test. For quantification, we used 2-OCT4 and 1-OCT4 bands for the *LIN28B* nucleosome, or 1-OCT4 and input nucleosome bands for the *LIN28B*-1M nucleosome (see Methods and Supplementary Table 3). **c**, View of the nucleosome entry–exit site showing the interaction of OCT4-POU_S with the histone H3 N-terminal and the histone H2A C-terminal tails. **d**, Electrostatic potential surface map of OCT4-POU_S interacting with the positively charged histone H3 tail. **e**, Representative native gel electrophoresis showing OCT4 binding to the *LIN28B* or *LIN28B*-H3K27ac nucleosomes (left). The asterisks are as in panel **b**, with the green asterisk marking 3-OCT4 (three OCT4 molecules bound). Quantification of OCT4 binding to H3K27ac relative to unmodified nucleosomes is also shown (right); 1st, 2nd or 3rd OCT4 (horizontal axis) indicates binding of the 1st, 2nd or 3rd molecule of OCT4, respectively. Data shown are mean ± s.e.m., *n* = 4 independent experiments; **P = 0.008 and ***P = 0.004, one-sided Student's *t*-test. **f**, Quantification of Mnl I digestion of free or OCT4-bound nucleosomes, unmodified or with H3K27ac. The *y* axis shows the intensity of nucleosome bands after digestion, normalized to input. Data shown are mean ± s.e.m., *n* = 4 independent experiments. Representative gels are shown in Extended Data Fig. 6d. **g**, Quantification of sequencing of MNase I-digested OCT4-bound nucleosomes, unmodified or with H3K27ac. The *x* axis shows the position of the first base pair relative to the most abundant position (0 as observed in the structure). Data are mean and spread of two independent experiments. A more detailed representation is shown in Extended Data Fig. 6f.

binding of the second and third OCT4 was increased with H3K27ac nucleosomes compared with unmodified nucleosomes (Fig. 3e). Deacetylation of H3K27ac abrogated the increased binding of the second and third OCT4, resulting in levels comparable with the unmodified nucleosome (Extended Data Fig. 6b,c). Thus, cooperative OCT4 binding to OBS2/3 is increased by H3K27ac.

This finding prompted us to examine whether the interactions of OCT4 with histone H3 contribute to DNA positioning by OCT4 in the *LIN28B* nucleosome. We first developed an assay to directly assess DNA positioning, taking advantage of an endogenous Mnl I restriction site between OBS1 and OBS2 (Fig. 1b); this site should be accessible to Mnl I when the *LIN28B* DNA is positioned as in our OCT4-bound nucleosome structure (Fig. 3c). Using *LIN28B* nucleosomes, we only observed partial digestion by Mnl I (Fig. 3f and Extended Data Fig. 6d), which is consistent with *LIN28B* DNA adopting multiple conformations on the

nucleosome. By contrast, the Mnl I restriction site was fully accessible in OCT4-bound nucleosomes, indicating that OCT4 binding stabilizes the nucleosomal DNA in a conformation in which the Mnl I site is exposed (Fig. 3f and Extended Data Fig. 6d). H3K27ac did not alter the sensitivity of nucleosomes alone to Mnl I digestion (Fig. 3f and Extended Data Fig. 6d); however, OCT4-bound H3K27ac nucleosomes showed higher protection from Mnl I digestion than OCT4-bound unmodified nucleosomes (Fig. 3f and Extended Data Fig. 6d). These results suggest that OCT4 induces an inward movement of the DNA on the H3K27ac nucleosome compared with the unmodified nucleosome. Such movement might be induced by the loss of electrostatic interaction between the H3K27 residue and the OCT4 acidic patch, and it would probably have limited range, until OCT4 gets close to the nucleosome; further DNA movement would require DNA unwrapping or OCT4 dissociation. The DNA movement would increase exposure of

OBS2/3, leading to higher binding of second and third OCT4 (Fig. 3a). To test this hypothesis, we digested OCT4-bound nucleosomes with MNase and sequenced the protected DNA (Extended Data Fig. 6e). We found that 70% of OCT4-bound unmodified nucleosomes were in a defined position, in agreement with our structural and biochemical data (Fig. 3g). By contrast, OCT4-bound H3K27ac nucleosomes were less well positioned, with a major species (20%) shifted by 1 bp inwards (Fig. 3g and Extended Data Fig. 6f).

To mimic the changes in the DNA positioning caused by OCT4 on H3K27ac nucleosomes, we moved OBS2/3 by 1 bp (*LIN28B*-OSO+1) or 2 bp (*LIN28B*-OSO+2) relative to OBS1. Modelling revealed that inward sliding of DNA for 1 bp would expose the binding site for OCT4-POU$_S$ at both OBS2 and OBS3, thus changing the interaction at OBS2 from OCT4-POU$_{HD}$ to OCT4-POU$_S$ (Extended Data Fig. 6g). We used unmodified nucleosomes bearing those constructs to test OCT4 binding and observed increased binding to OBS2/3 with the *LIN28B*-OSO+1 construct compared with *LIN28B* (Extended Data Fig. 6h). The *LIN28B*-OSO+2 construct showed binding of OCT4 comparable with *LIN28B* (Extended Data Fig. 6i). These data show that OCT4 binds to *LIN28B*-OSO+1 in a manner similar to its binding to the H3K27ac *LIN28B* nucleosome and support our conclusion that DNA movement of approximately 1 bp on OCT4-bound H3K27ac nucleosomes increases binding to internal sites. To test this model further, we examined the binding of OCT4 to H3K27ac *LIN28B*-OSO+1 nucleosomes, which would mimic a +2 bp movement (Extended Data Fig. 6j), and observed that acetylation of H3K27 in the *LIN28B*-OSO+1 nucleosome abrogated the increased binding of OCT4 to *LIN28B*-OSO+1 relative to the *LIN28B* nucleosome.

H3K27 methylation is a silent mark that would increase the bulkiness of the Lys residue, which could affect the interaction between the histone H3 tail and the OCT4 acidic patch. To test whether H3K27 methylation modulates OCT4 binding, we assembled nucleosomes with H3K27me3 and examined binding of OCT4 by native gel electrophoresis. We found that H3K27me3 did not significantly change OCT4 binding (Extended Data Fig. 7a). Consistent with that observation, we did not observe change in DNA positioning of OCT4-bound H3K27me3 nucleosomes by MnlI digestion and MNase sequencing compared with unmodified nucleosomes (Extended Data Figs. 6e and 7b,c).

## H3K27ac enhances OCT4–SOX2 cooperativity

In the *LIN28B* sequence, the SOX2-binding site is located between OBS2 and OBS3, forming a composite site with the latter. In our OCT4-bound nucleosome structure, the SOX2-binding site faces outwards, and modelling showed that SOX2 could bind to that site, with minor clashes with H2A (Extended Data Fig. 7d). Thus, OCT4 binding to the *LIN28B* nucleosome should facilitate SOX2 binding, by stabilizing the exposure of its binding site. We tested this hypothesis using native gel electrophoresis and observed that SOX2 could bind more efficiently to the OCT4-bound *LIN28B* nucleosome than to the *LIN28B* nucleosome alone (Extended Data Fig. 7e–g).

The inward DNA movement caused by OCT4 binding to the H3K27ac nucleosome would further increase exposure of the SOX2 site and alleviate the small clash between SOX2 and histone H2A (Extended Data Fig. 7d). Indeed, we found that SOX2 was able to bind better to OCT4-bound H3K27ac nucleosomes than to OCT4-bound unmodified nucleosomes (Fig. 4a). To validate our finding, we moved the SOX2-binding site to be 5 bp closer to OBS2, which would reduce its exposure in OCT4-bound nucleosomes but not in unbound nucleosomes that can slide. We observed that this shifting of the SOX2-binding site strongly reduced binding of SOX2 to OCT4-bound H3K27ac nucleosomes, indicating that OCT4 binding to OBS1 determines DNA positioning and OCT4–SOX2 cooperativity (Fig. 4b). Of note, SOX2 binding to free H3K27ac nucleosomes was not affected when its binding site was shifted by 5 bp, showing that DNA slides on free nucleosome, transiently exposing the SOX2-binding site and

allowing its binding. By contrast, when OCT4 is bound to the H3K27ac nucleosomes, DNA is positioned and cooperative binding with SOX2 is determined by the distance between OBS1 and the SOX2-binding site.

## OCT4 binding to the n*MATN1* nucleosome

To investigate whether our findings with the *LIN28B* nucleosome apply to other human DNA sequences, we assembled nucleosomes with a 186-bp-long DNA from the regulatory region n*MATN1* (ref. 3) (Extended Data Fig. 8a), which contains multiple OCT4-binding motifs. The initial cryo-EM reconstructions of the n*MATN1* nucleosome in complex with OCT4 showed a density near the linker DNA, similar to the density of OCT4 bound to *LIN28B* DNA (Extended Data Fig. 8b–g and Supplementary Table 1). Focused classification and refinements improved the resolution to 2.3 Å in the nucleosome portion (Extended Data Fig. 8c–f) and to 8.1 Å for an approximately 20-kDa OCT4 region bound to the linker DNA (Extended Data Fig. 9a–d). We performed MNase sequencing to determine the position of the n*MATN1* DNA on the OCT4-bound nucleosome and combined that information with the cryo-EM map to build a model for OCT4 bound to the n*MATN1* nucleosome (Extended Data Fig. 9d–g).

Our structural and MNase sequencing data revealed that, despite the presence of multiple OCT4 motifs in the n*MATN1* DNA, OCT4 predominantly binds to one binding site in the linker DNA, near the nucleosome entry–exit site (mOBS1) (Fig. 5a and Extended Data Fig. 9h), a position overall similar to that in the *LIN28B* nucleosome structure. Both DNA-binding domains of OCT4 engage the n*MATN1* nucleosome; whereas OCT4-POU$_{HD}$ is bound close to the nucleosome dyad, OCT4-POU$_S$ is located distally from the nucleosome (Fig. 5a,b). This arrangement differs from that with the *LIN28B* nucleosome, in which OCT4-POU$_S$ was bound to the linker DNA, close to the nucleosome dyad, whereas OCT4-POU$_{HD}$ was located distally from the nucleosome (Fig. 1a and Extended Data Fig. 2h). Nevertheless, despite its distal position relative to the n*MATN1* nucleosome, OCT4-POU$_S$ interacts with the histone H3 tail via a smaller acidic patch of OCT4 formed by the side chains of helices 4 and 5 (Fig. 5a,b and Extended Data Fig. 9i). Of note, the H3 tail from the entry–exit site opposite to the OCT4-binding side interacts with OCT4 (Fig. 5a,b and Extended Data Fig. 9i), in contrast to the *LIN28B* nucleosome.

OCT4 interaction with the H3 tail in the n*MATN1* nucleosome prompted us to test whether H3K27 modifications also modulate histone cooperativity on this human sequence, as it does for the *LIN28B* sequence. We found that both H3K27ac and H3K27me3 modifications increased binding of the second and especially the third OCT4 (Fig. 5c,d).

Together, our biochemical and structural data reveal the mechanism for cooperativity of OCT4 and SOX2. OCT4 binding to OBS1 on *LIN28B* or n*MATN1* nucleosomes stabilizes the positioning of nucleosomal DNA, to expose internal TF-binding sites, thereby facilitating binding of additional OCT4 and of SOX2. The internal binding sites are distant from OBS1 (Fig 3a and Extended Data Fig. 7d), pointing to an allosteric mechanism for TF cooperative binding, mediated by DNA positioning on the nucleosome. Moreover, H3K27 modifications can modulate cooperativity of OCT4 and downstream factors, such as other OCT4 molecules or SOX2, by altering the interaction between the OCT4 acidic patch and the H3 tail, which in turn affects the positioning of the nucleosomal DNA and exposure of internal binding sites for downstream factors.

## Discussion

Our cryo-EM structures captured OCT4 bound to nucleosomes assembled with endogenous *LIN28B* and n*MATN1* DNA and unveiled previously unknown OCT4 interactions with histones. A previous structure of OCT4 bound to engineered nucleosomes did not reveal interactions

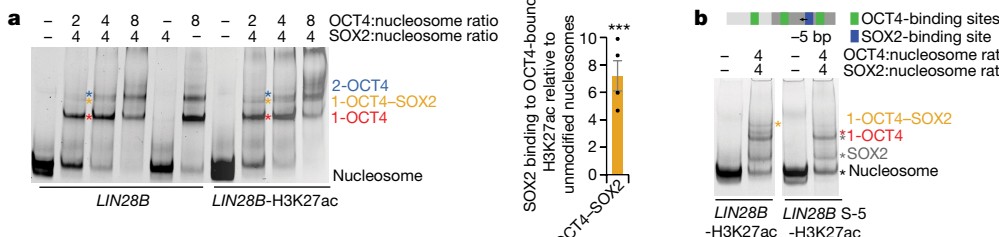

**Fig. 4 | Histone modifications modulate OCT4 and SOX2 cooperativity.**
**a**, Representative native gel electrophoresis showing OCT4 and SOX2 binding to *LIN28B* or *LIN28B*-H3K27ac nucleosomes (left). The coloured asterisks indicate molecules bound to the nucleosome: 1-OCT4 is in red, 1-OCT4 and 1-SOX2 are in orange and 2-OCT4 is in blue. Quantification of SOX2 binding to the OCT4-bound *LIN28B*-H3K27ac nucleosome relative to OCT4-bound *LIN28B* (right). Data shown as mean ± s.e.m., *n* = 4 independent experiments;

***P* = 0.0009, one-sided Student's *t*-test. **b**, Representative native gel electrophoresis (*n* = 2) showing OCT4 and SOX2 binding to *LIN28B*-H3K27ac nucleosomes and *LIN28B*-H3K27ac with the SOX2-binding site moved by 5 bp (arrow). The coloured asterisks indicate molecules bound to the nucleosome: the nucleosome is in black, 1-OCT4 is in red, 1-OCT4–SOX2 is in orange, and SOX2 is in grey.

with histones[21], but those nucleosomes contained the strong 601 positioning sequence[39] with the OCT4-binding site inserted in a position that could have prevented interactions with H3 and H2A tails. Our data indicate that proper positioning of the DNA-binding site for TFs on the nucleosome is required for specific interactions and for the formation of a stable OCT4–nucleosome complex. Our findings support a model in which initial binding of OCT4 to a partially exposed motif on the nucleosomal DNA leads to transient complexes that undergo DNA sliding to achieve stable OCT4 binding via its two DNA-binding domains (Extended Data Fig. 9j,k). This model is consistent with recent in vivo data showing that pioneer TFs bind preferentially next to nucleosomes[40].

Our findings show that a pioneer TF can directly alter the chromatin environment by stabilizing DNA on the nucleosome. DNA sliding on the nucleosome can occur spontaneously or be facilitated by chromatin remodelling complexes. In fact, OCT4 and other TFs can recruit chromatin remodelling complexes[41–45], which might facilitate nucleosome sliding to properly position DNA-binding motifs. The chromatin remodeller

BRG1 is required for OCT4 binding to a subset of gene regulatory elements in cells[13], and inhibition of the catalytic activity of BRG1 reduces the amount of already bound OCT4 at these elements in vivo[25], implying that chromatin remodellers support OCT4 by properly positioning nucleosomes at those specific locations. Our findings suggest that at other sites, OCT4 binding itself could directly position nucleosomal DNA and alter the accessibility of sites for downstream factors.

Of note, we observed that OCT4 alters the conformation of the N-terminal tail of histone H4, affecting internucleosome interactions and promoting chromatin decompaction. Recently, the SOX11 DNA-binding domain was proposed to affect the H4 tail position[46], but such mode of H4 regulation would limit TF binding to restricted regions on the nucleosome, where the DNA-binding domain would directly clash with the H4 tail. By contrast, our data reveal that the interaction between OCT4 and the histone H4 tail involves the disordered activation domain of OCT4 and takes place 70 Å away from the site where its DNA-binding domains interact with the nucleosome, indicating that interaction with the H4 tail does not depend on the

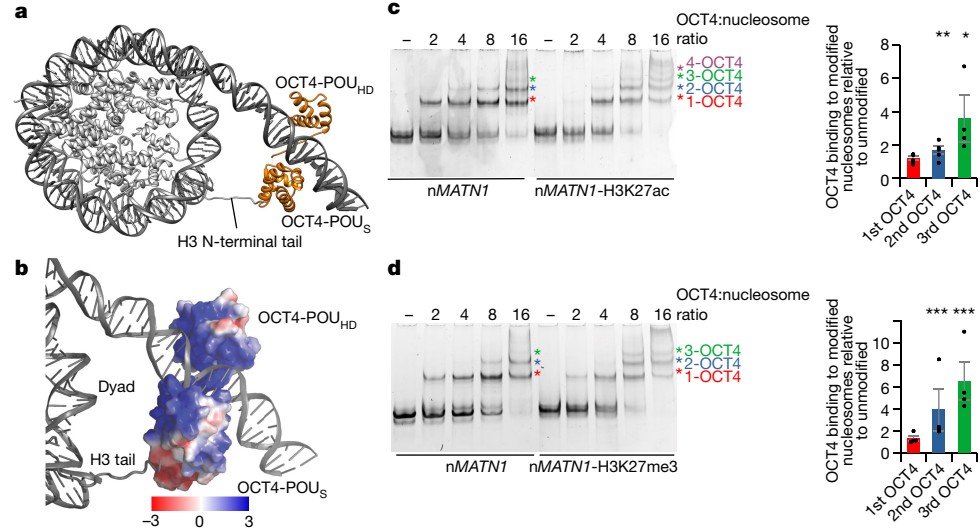

**Fig. 5 | Histone modifications modulate OCT4 binding to the n*MATN1* nucleosome. a**, Structural model of human OCT4 (orange) bound to a nucleosome (grey) assembled with a 186-bp DNA fragment from the n*MATN1* regulatory element. **b**, Surface model showing the electrostatic potential of the OCT4-POU_S domain and the positively charged histone H3 tail. **c**, Representative native gel electrophoresis showing OCT4 binding to unmodified or n*MATN1*-H3K27ac nucleosomes (left). The coloured asterisks mark molecules bound to the nucleosome: 1-OCT4 is in red, 2-OCT4 is in blue, 3-OCT4 is in green and 4-OCT4 is in purple. Quantification of OCT4 binding to n*MATN1*-H3K27ac

relative to unmodified nucleosomes (right), using bands marked with asterisks. Data shown are mean ± s.e.m. of four independent experiments; **P* = 0.02 and ***P* = 0.008, one-sided Student's *t*-test. **d**, Representative native gel electrophoresis showing OCT4 binding to unmodified or n*MATN1*-H3K27me3 nucleosomes (left). The coloured asterisks are as in panel **c**. Quantification of OCT4 binding to n*MATN1*-H3K27me3 relative to unmodified (right), using bands marked with asterisks. Data shown are mean ± s.e.m. of four independent experiments; ****P* = 0.0007 (2-OCT4) and ****P* = 0.0004 (3-OCT4), one-sided Student's *t*-test.

location of the OCT4-binding site. In agreement with our findings on OCT4, recent work has suggested that the activation domain of FOXA1 binds to histones and that this is required for FOXA1 to open chromatin, although the mechanism remains elusive[47].

Perhaps our most consequential finding is that TF binding and cooperativity can be regulated by histone modifications. Of note, our data show that H3K27 modifications did not affect the binding of the first OCT4 to the *LIN28B* or n*MATN1* nucleosome, but it altered the cooperative binding of additional OCT4 or of SOX2 to nucleosomal internal sites. These findings are consistent with previous in vivo data correlating OCT4-binding sites with H3K27ac[26,27,29]. However, positive correlation with histone marks has not been observed for FOXA2 and GATA4 (ref. 27), suggesting that not all TFs might be affected by epigenetic marks. In conclusion, our findings suggest that the pre-existing epigenetic landscape could tune pioneer TF activity.

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

# Methods

## Protein expression, mutagenesis and purification

*Xenopus laevis* histones for nucleosome assembly were overexpressed in the *Escherichia coli* BL21(DE3) pLysS strain and purified from inclusion body as previously described[48].

The cells were grown in LB medium at 37 °C and induced with 1 mM IPTG when $OD_{600}$ reached 0.6. After 3 h of expression, the cells were pelleted down, resuspended in lysis buffer (50 mM Tris-HCl (pH 7.5), 150 mM NaCl, 1 mM EDTA, 1 mM DTT and 0.1 mM PMSF) and frozen. Later, the frozen cells were thawed and sonicated. The pellet containing inclusion bodies was recovered by centrifugation at 5,000 rpm for 20 min at 4 °C. The inclusion body pellet was washed three times with lysis buffer containing 1% Triton X-100, followed by two washes with lysis buffer without Triton X-100.

Each histone protein was extracted from the purified inclusion body pellet in a buffer containing 50 mM Tris (pH 7.5), 2 M NaCl, 6 M guanidine hydrochloride and 1 mM DTT for overnight at room temperature. Any insoluble components were removed by centrifugation. Proteins making histone pairs (H2A–H2B and H3–H4) were combined in equimolar ratios and dialysed two times in 1 l of refolding buffer (25 mM HEPES/NaOH (pH 7.5), 2 M NaCl and 1 mM DTT) at 4 °C. Any precipitate was removed by centrifugation for 20 min at 13,000 rpm at 4 °C. The soluble histone pairs were further purified via cation-exchange chromatography in batch (SP Sepharose Fast Flow resin). The samples were diluted fourfold with buffer without salt (25 mM HEPES/NaOH (pH 7.5) and 1 mM DTT) and bound to the resin for 30 min. The resin was extensively washed with 500 mM salt buffer in batch (25 mM HEPES/NaOH (pH 7.5), 500 mM NaCl and 1 mM DTT) and loaded onto a disposable column. On the column, the resin was washed, and pure proteins were eluted with 25 mM HEPES/NaOH (pH 7.5), 2 M NaCl and 1 mM DTT. Soluble histone pairs were concentrated and purified on a Superdex S200 size-exclusion column (GE) equilibrated in 25 mM HEPES/NaOH (pH 7.5), 2 M NaCl and 1 mM DTT. Clean protein fractions were pooled, concentrated and flash frozen.

For cryo-EM grid freezing of 'assembly 1' (see below), commercially available OCT4 from Abcam (ab 134876) was used. The protein (approximately 52 kDa) was fused with the herpes simplex virus VP16 transactivation domain at the N terminus and a 11R tag at the C terminus. For the 'assembly 2' for cryo-EM and all the other assays, His-tagged OCT4 (approximately 39 kDa) was expressed in a pET28 vector and purified under denaturing conditions from inclusion body using Talon affinity resins. To refold the OCT4 protein, the first overnight dialysis was carried out in 2 M urea, 50 mM HEPES (pH 7.5), 250 mM NaCl, 50 mM L-arginine and 2 mM DTT. Then, the second and third dialyses were carried out for 1 h in a buffer containing 50 mM HEPES (pH 7.5), 100 mM NaCl and 1 mM DTT.

All the OCT4 variants were generated using the inverse PCR strategy. Oligo primers used for mutagenesis were purchased from Integrated DNA Technology and are listed in the Supplementary Table 1. The inverse PCRs were set up in a total volume of 25 µl. After amplification, 10 µl of purified PCR product was incubated with 5 U of T4 PNK in 20 µl of 1× T4 DNA ligase buffer for 1 h at 37 °C. Of T4 DNA ligase, 200 U was added to the reaction and incubated for 1 h at room temperature. Finally, 10 U of Dpn I was added to the reaction and incubated for 1 h at 37 °C. From this mixture, 5 µl was used to transform the competent XL1-Blue *E. coli* cells. The clones were selected on kanamycin plates and were subsequently confirmed by sequencing.

## Histone octamer assembly and purification

Histone octamer purification was done using the standard protocol[48,49]. In brief, a 2.5-fold molar excess of the H2A–H2B dimer was mixed with the H3–H4 tetramer in the presence of buffer containing 2 M NaCl (25 mM HEPES (pH 7.5), 2 M NaCl and 1 mM DTT). After overnight incubation at 4 °C, the assembled octamer was separated from excess dimer using a Superdex S200 Increase 10/300 GL column on an AKTA FPLC system. The fractions were analysed on SDS–PAGE, pooled and concentrated for final nucleosome assembly.

## *LIN28B* 182-bp DNA amplification

A custom synthesized (Integrated DNA Technology) 162-bp *LIN28* genomic DNA[8] was cloned into the pDuet plasmid. To make the longer 182-bp *LIN28B* DNA fragment by PCR, two primers were designed so that each contained an extra 10 bases from the flanking genomic region of the canonical 162-bp *LIN28* fragment used in previous studies[8]. The DNA sequence for the 182-bp extended DNA used in this study is shown in Supplementary Table 1.

## Mutant *LIN28B* DNA

Custom synthesized 182-bp *LIN28B* DNA was purchased from Integrated DNA Technology with the following mutations in the three OCT4-binding sites:

*LIN28B*-1M: ATT AAC AT - GCGTCGAT
*LIN28B*-2M: ATT AAC AT - GCG GCT AT
*LIN28B*-3M: ATG CTG AAT - GCG GGT AA

The fragments were later PCR amplified to generate DNA for nucleosome assemblies.

## OCT4-binding DNA sequences from the human genome

The 186-bp n*MATN1* sequence was selected from the human genome (https://www.ncbi.nlm.nih.gov/genome/gdv) from the position GRCh38:1:30216402:30217024:1 on chromosome 1 (ref. 3). The DNA fragment was selected based on the presence of the following OCT4 motifs: ATGCTAAT, ATTAGCAT, ATTAACAT or ATGTTAAT. The 186-bp n*MATN1* sequence is shown in Supplementary Table 1.

## Nucleosome assembly

Nucleosome assembly was carried out using a 'double bag' dialysis method as previously described[50,51]. The histone octamer and nucleosomal DNA fragment were mixed in equimolar ratios in a buffer containing 50 mM HEPES (pH 7.5), 2 M NaCl and 2 mM DTT. The mixture was placed into a dialysis button made with a membrane with a cut-off of 3.5 kDa. The dialysis button was placed inside a dialysis bag (6–8-kDa cut-off membrane) filled with 50 ml of buffer containing 25 mM HEPES (pH 7.5), 2 M NaCl and 1 mM DTT. The dialysis bag was immersed into 1 l of buffer containing 25 mM HEPES (pH 7.5), 1 M NaCl and 1 mM DTT, and dialyzed overnight at 4 °C. The next day, the buffer was changed to 1 l of a buffer with 25 mM HEPES (pH 7.5) and 1 mM DTT, and dialysis was continued for 6–8 h. In the last step, the dialysis button was removed from the dialysis bag and dialysed overnight into a fresh buffer without any salt (50 mM HEPES (pH 7.5) and 1 mM DTT). The nucleosome assemblies were assessed on a 6% native PAGE using SYBR Gold staining.

## Assembly of modified nucleosomes

H3K27ac nucleosomes were assembled using the *LIN28B* DNA (unlabelled or Cy5-labelled) and histone octamer with H3K27ac modification (custom purchased from Epicypher). For H3K27me3 nucleosomes, the H3K27C-mutant histone was generated using site-directed mutagenesis and later expressed and purified from *E. coli*. The H3K27C-mutant histone thus obtained was trimethylated using the MLA protocol[52] and was purified using a PD10 column. This trimethylated H3 was used with other histones for octamer assembly. The purified H3K27me3 octamer was mixed with *LIN28B* and n*MATN1* DNA for the assembly of H3K27me3 nucleosomes.

## Nucleosome array assembly

A 1,022-bp genomic region from the *LIN28* genomic site was synthesized by DNA synthesis (Codex, Protein Technology Center, St Jude Children's Research Hospital). For nucleosome array reconstitution, the DNA

fragment was amplified to a larger scale by PCR. For the assembly, the DNA and histone octamer were mixed in a 1:5 ratio.

Nucleosome array reconstitution was carried out using the double bag dialysis salt dilution method described above (see 'Nucleosome assembly').

The synthesized genomic DNA sequence used for array assembly (the *LIN28B* 182-bp region shown in bold) is shown in Supplementary Table 1.

## Assembly of the nucleosome–OCT4 complex for cryo-EM grid freezing

**The *LIN28B* complex.** Equimolar mixture of histone octamer and *LIN28B* DNA (2 μM each) were mixed with 1 μM of OCT4 (ab134876, Abcam) and 3 μM of SOX2 (50 mM HEPES (pH 7.5), 2 M NaCl, 20% glycerol and 5 mM DTT). The assembly was carried out with four steps of buffer changes over 72 h. The buffer changes were carried out to dilute out the salt concentration from 2 M starting concentration to a final solvent condition of no salt. The three buffers, used for the assembly dialysis, contained 50 mM HEPES (pH 7.5), 2 mM DTT and varying NaCl concentrations of 2 M, 1 M and 0, respectively. After the assembly, the samples were centrifuged at 13,000 rpm for 10 min at 4 °C to remove any precipitates. Following this, the sample was concentrated using a 10 kDa Centricon to the concentrations needed for cryo-EM grid freezing (0.5–1 μg μl$^{-1}$).

The assemblies were checked on 6% native gels followed by native western blot analysis. For the detection of nucleosomes, OCT4 and SOX2, anti-H3, anti-OCT4 and anti-His antibodies were used, respectively (see the section 'Western blot detection' below).

**The n*MATN1* complex.** For the OCT4 bound to the n*MATN1* nucleosome, 1 μM of pre-assembled nucleosomes were mixed with 2 μM of His-tagged OCT4 (see above) and incubated at room temperature for 30 min. The sample was then transferred to ice until grid freezing.

## Restriction enzyme Mnl I digestion assays

For digestion of *LIN28B* nucleosomes, different dilutions of Mnl I (NEB) were made in the 1× CutSmart buffer (NEB). The digestion was carried out for 30 min at 25 °C. For the experiments involving OCT4 and OCT4 variants, the protein was incubated with nucleosomes at 25 °C for 5 min before the addition of Mnl I. After the addition of Mnl I, the samples were kept at 25 °C for 30 min. After the digestion, the samples were run on a 6% polyacrylamide gel to separate all the products and then imaged by SYBR Gold staining on a Typhoon scanner.

## Magnesium precipitation assay

*LIN28B* nucleosome samples were incubated in varying MgCl$_2$ concentrations for 10 min at 25 °C. The precipitated nucleosomes were separated from soluble nucleosomes by spinning at 10,000 rpm for 10 min at 25 °C. The same procedure was followed for nucleosome samples containing wild-type OCT4 and other variants. However, for the experiments done in the presence of OCT4 and OCT4 variants, the nucleosomes were first mixed with fivefold molar excess of OCT4 (or OCT variants) and kept at 25 °C for 5 min before any MgCl$_2$ addition.

## Binding assays

The binding assays with OCT4 were performed at 25 °C in 50 mM HEPES (pH 7.5), 200 mM KCl, 1 mM DTT and 0.005% NP-40. The binding assays involving both OCT4 and SOX2 were performed in 50 mM HEPES (pH 7.5) and 1 mM DTT. Typically, 20–40 nM of nucleosome was incubated with different amounts of proteins (OCT4, OCT4 variants and SOX2). For the OCT4-binding experiments, the reaction was incubated for 10 min. For binding involving both OCT4 and SOX2, the reaction was incubated for 10 min after the addition of OCT4, following which SOX2 was added and kept for an additional 5 min. The bound and unbound species were separated on a 5% or 6% native polyacrylamide gel and imaged for Cy5 fluorescence using a Typhoon scanner. In experiments

with nucleosomes without Cy5 label, SYBR Gold staining was used to visualize the gels.

## Analysis of gels

All the gels were analysed using Quantity One Basic version (Bio-Rad). The data were exported and analysed or plotted using Open Office Calc. All the bands were selected using boxes of the same size: 24 mm$^2$ for input nucleosome and 8 mm$^2$ for all other bands. The background correction was done separately for bands from each lane using boxes of identical size in the same lane.

## Analysis of Mnl I digestion of nucleosomes.
In the nucleosome-only experiment, after background correction, the signal from the nucleosome band from each concentration point was normalized to the signal from the nucleosome lane in the 0 Mnl I lane. For Mnl I digestion in the presence of OCT4 or its variants, the signal of the OCT4-bound band from each of the Mnl I concentration was background corrected and then normalized to the signal of the OCT4-bound band from the 0 Mnl I lane.

## Analysis of the Mg$^{2+}$ precipitation assays.
The relative compaction was calculated as the fraction of the precipitated nucleosomes. For this, the following formula was used: relative compaction = $S_0 - S_{obs}$. $S_0$ is the signal of the nucleosome band at the 0 Mg$^{2+}$ concentration normalized to 1, and $S_{obs}$ is the signal of all the soluble nucleosome bands normalized to the signal of nucleosomes at the 0 Mg$^{2+}$ concentration. For precipitation experiments in the presence of OCT4 or its variants, the signals from both the bound and the unbound nucleosomal species were summed to calculate the soluble nucleosomes.

## Analysis of OCT4 binding to wild-type *LIN28B* versus the *LIN28B*-1M mutant.
For binding to wild-type *LIN28B* nucleosomes, all bands were normalized to input nucleosome. For comparison, we used the following equation: binding to OBS2/3 = '2-OCT4'/('1-OCT4' + '2-OCT4'), where '2-OCT4' represents a nucleosome with two OCT4 bound (OBS1 + OBS2/3), and '1-OCT4' is a nucleosome with one OCT4 bound (OBS1). '1-OCT4' + '2-OCT4' represents input OCT4 bound to OBS1 nucleosomes, which are substrates for binding of the second OCT4. For binding to *LIN28B*-1M nucleosomes, we used the following equation: binding to OBS2/3 = '1-OCT4'/nucleosome, where nucleosome represents input nucleosomes.

## Analysis of SOX2 binding to wild-type *LIN28B*.
Binding of SOX2 to OCT-bound nucleosome was calculated as the fraction of SOX2 bound to the OCT4-bound *LIN28B* nucleosome: SOX2 = '1-OCT4–SOX2'/('OCT4' + '1-OCT4–SOX2'), where '1-OCT4–SOX2' represent nucleosomes with both OCT4 and SOX2 bound, and OCT4 represents OCT4-bound nucleosomes. '1-OCT4' + '1-OCT4–SOX2' represents input OCT4-bound nucleosomes, which are substrates for binding of the SOX2 to OCT4-bound nucleosomes. Binding of SOX2 to the *LIN28B* nucleosome is shown as a fraction of free *LIN28B* nucleosomes: SOX2 = SOX2/nucleosome, where SOX2 represents SOX2-bound nucleosomes and nucleosome represents input nucleosomes.

## Analysis of OCT4 binding to unmodified, H3K27ac and H3K27me3 nucleosomes.
For binding to modified nucleosomes, we used the following equations: 1st OCT4 = ('1-OCT4' + '2-OCT4' + '3-OCT4')/input nucleosome; 2nd OCT4 = ('2-OCT4' + '3-OCT4')/(input nucleosome); and 3rd OCT4 = '3-OCT4'/(input nucleosome), where 1st, 2nd or 3rd OCT4 indicates binding of the 1st, 2nd or 3rd molecule of OCT4, respectively, '1-OCT4' is a nucleosome with one OCT4 bound, '2-OCT4' is a nucleosome with two OCT4 bound, and '3-OCT4' is a nucleosome with three OCT4 bound. The quantification is shown as a ratio of modified nucleosomes to unmodified nucleosomes (1st OCT4 modified/1st OCT4 unmodified).

**Analysis of SOX2 binding to H3K27ac nucleosomes.** For binding to modified nucleosomes, we used the following equation: 'SOX2–OCT4' = '1-OCT4–SOX2'/('1-OCT4–SOX2' + '1-OCT4'), where 'SOX2–OCT4' represents SOX2 binding to nucleosome with one OCT4 bound, '1-OCT4' represents a nucleosome with one OCT4 bound, and '1-OCT4–SOX2' is a nucleosome with OCT4 and SOX2 bound. The quantification is shown as a ratio of modified nucleosomes to unmodified nucleosomes ('SOX2–OCT4' modified/'SOX2–OCT4' unmodified).

### Western blot detection
SDS–PAGE gels or native PAGE gels were transferred to a PVDF membrane and blocked in TBST (50 mM Tris/HCl (pH 7.5), 150 mM NaCl and 0.1% Tween-20) containing 5% milk for 1 h. Membranes were then incubated in primary antibody in TBST containing 5% milk for 1 h at room temperature. The membranes were washed three times for 5 min with TBST and incubated in secondary antibody for 1 h at room temperature. Membranes were washed three times (approximately 5 min each) with TBST before chemiluminescent detection. The following antibodies were used: anti-OCT4 antibody (1:2,000 dilution; ab109183, Abcam), horseradish peroxidase-conjugated anti-His antibody (1:3,000 dilution; R931-25, Invitrogen–Thermo Fisher), anti-H3 antibody (1:3,000 dilution; ab1791, Abcam) and anti-SOX2 antibody (1:2,000 dilution; ab92494, Abcam), horseradish peroxidase-conjugated anti-rabbit secondary antibody (1:2,000 dilution; 170-6515, Bio-Rad).

### MNase-seq
OCT4 was bound to unmodified, H3K27ac or H3K27me3 nucleosomes with *LIN28B* or n*MATN1* DNA (20 mM HEPES (pH 7.5), 50 mM KCl, 2.5 mM $MgCl_2$ and 5 mM $CaCl_2$) and digested by MNase (NEB) for 5 min at 25 °C. MNase digestion was terminated by 50 mM EDTA. Cleaved nucleosome was subjected to phenol/chloroform extraction followed by ethanol precipitation of nucleosomal DNA and used for library preparation. The sequencing library was prepared using the NEBNext Ultra II DNA Library Prep Kit following the manufacturer's manual. Amplification of the library for Illumina sequencing was performed by PCR using NEBNext Multiplex Oligos for the Illumina kit. Sequencing was pair ended with 100-bp length. Paired reads were merged and filtered by the length of reads between 144 bp and 146 bp and mapped to the *LIN28B* or n*MATN1* sequence with Qiagen CLC genomics Workbench 20 software.

### MiDAC purification
MiDAC was purified from 1.25 l of adherent Flp-In 293 T-REx (R78007, Thermo Fisher Scientific) cell lines stably transformed with the Flp-In expression vector carrying FLAG-ELMSAN1/MIDEAS. The cells were grown in DMEM media (Gibco) supplemented with 10% FBS, 100 μg ml⁻¹ hygromycin and induced for 24 h with 1 μg ml⁻¹ doxycycline (Thermo Fisher Scientific). Cells were harvested and lysed using the classical Dignam protocol[53]. The complex was isolated from the nuclear fraction using anti-FLAG M2 beads from Sigma-Aldrich. The nuclear fraction was mixed with washed FLAG M2 beads and incubated overnight at 4 °C. The next day, the beads were washed with wash buffer (20 mM HEPES (pH 7.9), 300 mM NaCl, 1.5 mM $MgCl_2$, 10% glycerol, 0.5 mM DTT and protease inhibitors (Sigma)) four times. The complex was eluted from the beads in the elution buffer (20 mM HEPES (pH 7.9), 100 mM NaCl, 1.5 mM $MgCl_2$, 0.5 mM DTT and protease inhibitors (Sigma)) after 30 min of incubation at 4 °C. This complex was flash frozen in liquid nitrogen and stored at −80 °C.

### Deacetylation of H3K27ac nucleosomes
H3K27ac nucleosomes were deacetylated by the human MiDAC deacetylase complex. The deacetylation reaction was carried out for 18 h at 25 °C in the following buffer: 50 mM HEPES (pH 7.5), 100 mM KCl and 0.2 mg ml⁻¹ BSA. A control parallel reaction containing H3K27ac nucleosomes, but no MiDAC, was also carried out under identical conditions. The extent of deacetylation was confirmed by western blot using anti-H3K27ac antibody.

### Negative-stain EM
For the experiment looking at array compaction, 20 nM of the *LIN28* array was mixed with $MgCl_2$ to a final [$Mg^{2+}$] of 3 mM. For analysis of the effect of wild-type and ΔN OCT4 proteins, 70 nM (wild type) and 100 nM (ΔN) proteins were used with the mixture of array and $MgCl_2$.

After approximately 10–15 min of incubation at 25 °C, 3 μl of the sample was added to Lassey carbon or quantifoil grids for 1 min, blotted dry and stained. For staining, four separate drops (approximately 40 μl) of uranyl acetate or uranyl formate were added to a parafilm strip. The grid was briefly brought into contact with the stain for the first three drops before quick blotting. The last drop of stain was kept in contact with the grid for 1 min before the final blot drying.

The dried grids were imaged on a Talos L 120C microscope (Thermo Fisher Scientific) at the cryo-EM facility at St Jude Children's Research Hospital. Several images were acquired at ×73,000–92,000 magnification from regions showing good particle distribution. Specifically, a magnification of ×73,000 was used for experiments involving $Mg^{2+}$ compacted arrays in the absence or presence of OCT4; for the experiment with the ΔN variant of OCT4, a magnification of ×92,000 was used. The pixel size was 1.94 Å (73,000) to 1.54 Å (92,000) per pixel on the object scale. The images were later analysed using the ImageJ software after matching the scale from the EM images.

### Negative-stain image analysis
Several particles were picked using RELION ($n$ = 450 for arrays in 3 mM $MgCl_2$, $n$ = 262 for arrays in 3 mM $MgCl_2$ with OCT4 and $n$ = 307 for arrays in 3 mM $MgCl_2$ with the ΔN variant of OCT4). For particle picking, the images from the microscope were binned twofold in RELION and saved as 400 pixel × 400 pixel tiff files, which were later analysed using the ImageJ software[54]. First, the particles were encircled using the free-form selection tool in ImageJ. Later, the 'set scale' tool in ImageJ was used to set the size of the pixel in the image to 0.4 nm (pixel size of 0.2 nm at ×73,000 magnification multiplied by 2 for binning in RELION). The particle sizes were measured using the image analyser option in ImageJ and plotted.

### Cryo-EM grid preparation and data collection
For cryo-EM of the OCT4-bound *LIN28B* nucleosome structure, we assembled an OCT4–SOX2–nucleosome complex as described. The sample was concentrated to 0.25 mg ml⁻¹ for the cryo-EM grid. To avoid the extensive aggregation of the complex sample on the cryo-EM grid, OCT4 and SOX2 were mixed with nucleosomes in a 0.5:1 ratio during the assembly. The OCT4-bound n*MATN1* nucleosome was assembled as described with a 2:1 ratio of OCT4 to nucleosome. Of the complex sample, 3 μl was applied to a freshly glow-discharged Quantifoil R2/1 holey carbon grid. The humidity in the chamber was kept at 95% and the temperature at +10 °C. After 5 s of blotting time, grids were plunge-frozen in liquid ethane using a FEI Vitrobot automatic plunge freezer.

For the *LIN28B* nucleosome and the OCT4-bound *LIN28B* nucleosome, electron micrographs were recorded on FEI Titan Krios at 300 kV with a Gatan Summit K3 electron detector using SerialEM[55] (approximately 6,000 and approximately 11,000 micrographs, respectively) at the Cryo-EM facility at St. Jude Childrens's Research Hospital. Image pixel size was 1.06 Å per pixel on the object scale. Data were collected in a defocus range of 7,000–30,000 Å with a total exposure of 90 e⁻ Å⁻². Fifty frames were collected and aligned with the MotionCorr2 software using a dose filter[56,57]. The contrast transfer function parameters were determined using CTFFIND4 (ref. 58). For the OCT4-bound n*MATN1* nucleosome, the data were recorded on the FEI Titan Krios at 300 kV with a Falcon 4 electron detector using EPU (approximately 35,000 micrographs) at the Cryo-EM facility at the Dubochet Center for Imaging (DCI) at EPFL and UNIL. Data were collected in a defocus range of

7,000–25,000 Å. Image pixel size was 0.83 Å per pixel on the object scale.

Several thousand particles were manually picked and used for training and automatic particle picking in Cryolo[59]. Particles were windowed and 2D class averages were generated with the RELION software package[60]. Inconsistent class averages were removed from further data analysis. The initial reference was filtered to 40 Å in RELION. C1 symmetry was applied during refinements for all classes. Particles were split into two datasets and refined independently, and the resolution was determined using the 0.143 cut-off (RELION auto-refine option). All maps were filtered to resolution using RELION with a B-factor determined by RELION.

Initial 3D refinement was done with 2,600,000 particles. To improve the resolution of this flexible assembly, we used focused classification followed by focused local search refinements. Nucleosomes were refined to 2.8 Å. Density modification in Phenix improved the map to 2.5 Å (ref. 61). OCT4 bound to DNA (30 kDa) was refined to 4.2 Å using a subset of 65,000 particles after extensive sorting. Using density modification in Phenix, we improved resolution and the appearance of this density to 3.9 Å. The maps have extensive overlapping densities that we used to assemble the composite map and model. The *LIN28B* nucleosome sample contained 1,000,000 particles, which were refined to 3.1 Å, and improved with density modification to 2.8 Å.

For the second dataset, we collected 1,400 images, yielding 68,000 nucleosomal particles, which refined to 3.7 Å. Classification revealed that approximately 21,000 particles had OCT4 bound, which refined to 4.2 Å.

Molecular models were built using Coot[62]. The model of the nucleosome (Protein Data Bank (PDB): 6WZ5)[63] was refined into the cryo-EM map in PHENIX[64]. The model of the OCT4 bound to DNA (PDB: 3L1P)[19] were rigid-body placed using PHENIX, manually adjusted and rebuilt in Coot and refined in Phenix. Visualization of all cryo-EM maps was done with Chimera[65].

## Reporting summary

Further information on research design is available in the Nature Portfolio Reporting Summary linked to this article.

## Data availability

EM density maps and models have been deposited in the Electron Microscopy Data Bank and PDB under the following accession codes: for OCT4 bound to the *LIN28B* nucleosome, PDB 8G8G was built using maps EMD-29855 (all particles), EMD-29850 (H3 tail subset), EMD-29852 (H2A tail subset), EMD-29854 (H4 tail A subset), EMD-29854 (H4 tail B subset) and EMD-29846 (OCT4 focus; PDB 8G8E). For OCT4 bound to the n*MATN1* nucleosome, EMD-29837 and PDB 8G86 (nucleosome focus); EMD-29841 and PDB 8G87 (OCT4 focus); EMD-29843 and PDB 8G88 (conformation 1); and EMD-29845 and PDB 8G8B (conformation 2).

All other data supporting the findings of this study are available within the article and its Supplementary information files.

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

**Acknowledgements** We thank Cryo-EM Center members at St. Jude Children's Research Hospital for support with grid screening, especially A. Myasnikov and L. Tang for support with data collection of the *LIN28B* nucleosome and the OCT4-bound *LIN28B* nucleosome; A Myasnikov and B. Beckert from the Dubochet Center for Imaging (DCI) at EPFL and UNIL for data collection of OCT4 bound to the n*MATN1* nucleosome; H.-M. Herz and Y. Sedkov for FLAG–ELMSAN1 HEK293T cell line and support with MiDAC purifications; I. Chen for critical reading and comments; and Z. Luo for help with making Extended Data Fig. 9j,k. Work in the Halic laboratory is funded by St. Jude Children's Research Hospital, the American Lebanese Syrian Associated Charities and the US NIH awards 1R01GM135599-01 and 1R01GM141694-01.

**Author contributions** K.K.S. and M.H. designed the experiments. K.K.S. performed the biochemical experiments and electron microscopy. S.B. built the models. Y.D. performed the MNase-seq experiments. D.M. purified the MiDAC complex. K.K.S., S.B., Y.D. and M.H. analysed the data. K.K.S., S.B. and M.H. wrote the paper.

**Competing interests** The authors declare no competing interests.

**Additional information**
**Correspondence and requests for materials** should be addressed to Mario Halic.

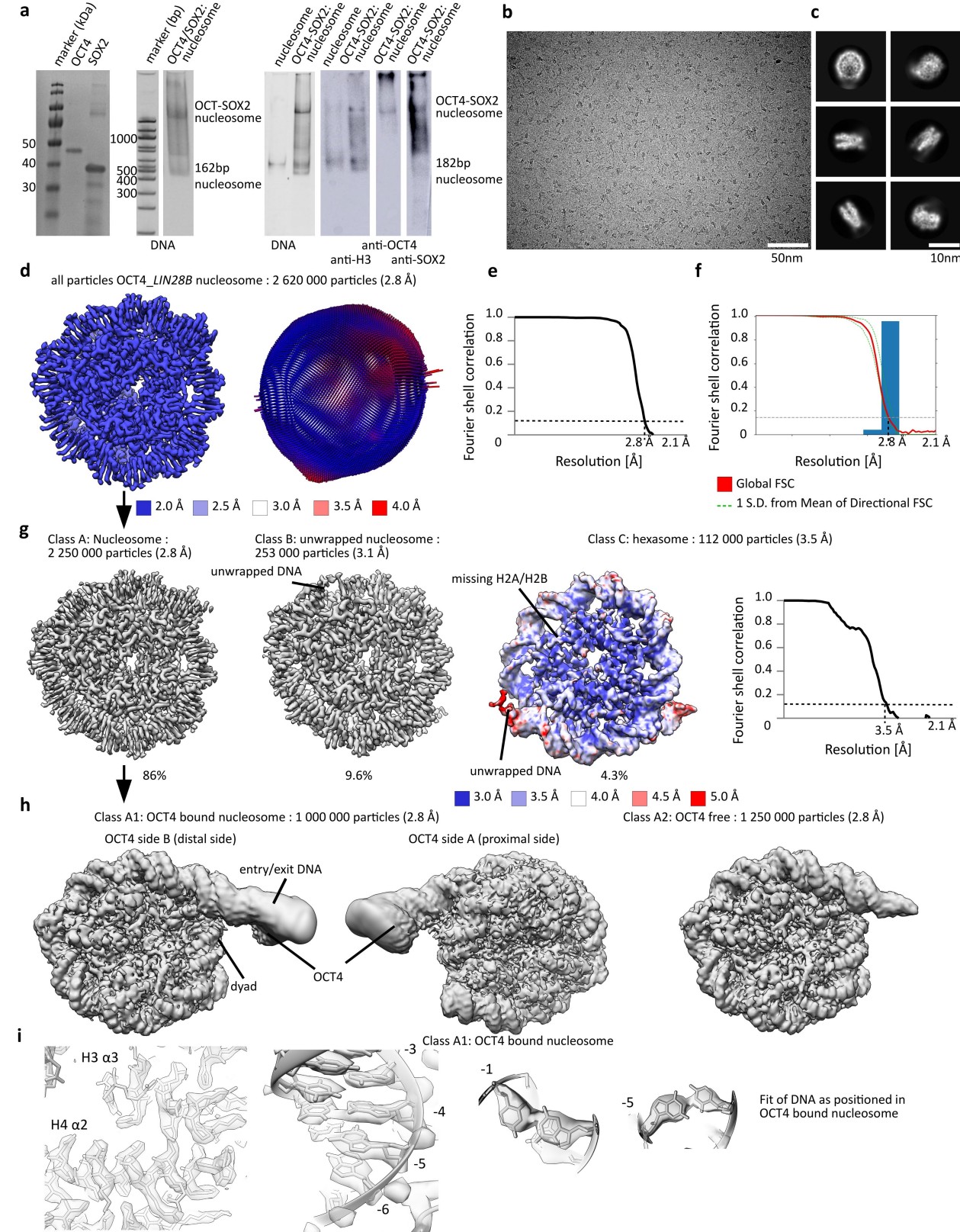

**Extended Data Fig. 1** | See next page for caption.

**Extended Data Fig. 1 | Assembly and cryo-EM of OCT4 bound to *LIN28B* nucleosomes. a)** SDS-PAGE showing purification of OCT4 and SOX2 and the assembly of the OCT4-SOX2-nucleosome complex. From left: A SDS gel showing purification of OCT4 and SOX2 used in the experiments; a native gel stained for DNA showing the assembly of the OCT4_SOX2_nucleosome complex; western blots with anti-H3 antibody, anti-OCT4 antibody and anti-His antibody (SOX2). Each of these experiments have been repeated > 3 times. See Supplementary Fig. 1 for original uncropped images. **b)** Representative cryo-EM micrograph from a set of 11 000 micrographs collected with Titan Krios electron microscope at 300 keV. Nucleosome particles in multiple orientations are visible. **c)** Representative 2D class averages showing nucleosomes. **d)** Cryo-EM map of nucleosome from the entire dataset, refined to 2.8 Å. The map is colored by local resolution. The model of the nucleosome (PDB: 6WZ5) was refined into the cryo-EM map. Angular distribution for nucleosome is shown on the right. **e)** Fourier shell correlation (FSC) curve showing the resolution of the map in d). **f)** Directional FSC plot showing uniform resolution in all directions. **g)** Classification of the data in b) resulted in three major classes of nucleosomes and nucleosome like particles (nucleosome, unwrapped nucleosome and hexasome). Hexasome map is colored by local resolution and the FSC curve is shown on the right. Number of particles corresponding to each class is indicated. **h)** Classification of the nucleosome subset from g). Classification revealed two classes, nucleosome and nucleosome with bound OCT4. We did not observe density for SOX2. **i)** Left: the representative region showing map quality and fit of the model is shown for the nucleosome with bound OCT4. Right: bases in the DNA are well resolved.

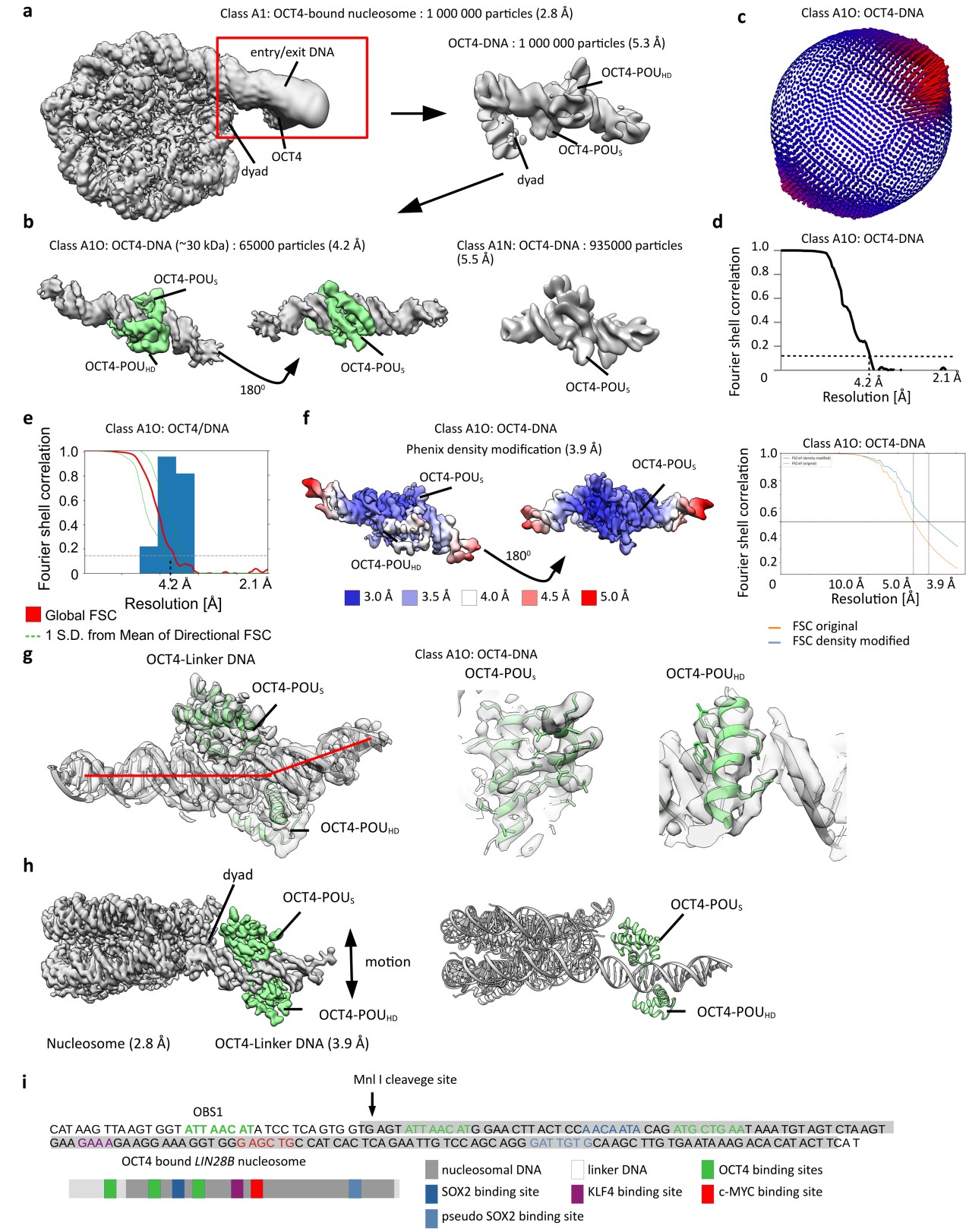

**a** Class A1: OCT4-bound nucleosome : 1 000 000 particles (2.8 Å)

entry/exit DNA

OCT4-DNA : 1 000 000 particles (5.3 Å)

OCT4-POU_HD

OCT4-POU_S

OCT4

dyad

dyad

**c** Class A1O: OCT4-DNA

**b** Class A1O: OCT4-DNA (~30 kDa) : 65000 particles (4.2 Å)

OCT4-POU_S

OCT4-POU_HD

180°

OCT4-POU_S

Class A1N: OCT4-DNA : 935000 particles (5.5 Å)

OCT4-POU_S

**d** Class A1O: OCT4-DNA

**e** Class A1O: OCT4/DNA

■ Global FSC

--- 1 S.D. from Mean of Directional FSC

**f** Class A1O: OCT4-DNA
Phenix density modification (3.9 Å)

OCT4-POU_S

OCT4-POU_HD

180°

OCT4-POU_S

3.0 Å   3.5 Å   4.0 Å   4.5 Å   5.0 Å

Class A1O: OCT4-DNA

FSC original
FSC density modified

**g** OCT4-Linker DNA

OCT4-POU_S

OCT4-POU_HD

Class A1O: OCT4-DNA
OCT4-POU_S         OCT4-POU_HD

**h**

dyad

OCT4-POU_S

motion

OCT4-POU_HD

Nucleosome (2.8 Å)   OCT4-Linker DNA (3.9 Å)

OCT4-POU_S

OCT4-POU_HD

**i**

Mnl I cleavege site

OBS1

CAT AAG TTA AGT GGT **ATT AAC AT**A TCC TCA GTG GTG AGT ATT AAC ATG GAA CTT ACT CCA ACA ATA CAG ATG CTG AAT AAA TGT AGT CTA AGT
GAA GAA AGA AGG AAA GGT GGG AGC TGC CAT CAC TCA GAA TTG TCC AGC AGG GAT TGT GCA AGC TTG TGA ATA AAG ACA CAT ACT TCA T

OCT4 bound *LIN28B* nucleosome

■ nucleosomal DNA   ☐ linker DNA   ■ OCT4 binding sites
■ SOX2 binding site   ■ KLF4 binding site   ■ c-MYC binding site
■ pseudo SOX2 binding site

**Extended Data Fig. 2** | See next page for caption.

**Extended Data Fig. 2 | Classification of the OCT4-nucleosome complex.**
**a)** Focused refinement of the OCT4 density from the OCT4-nucleosome (left) complex improved the resolution in the OCT4 bound region to 5.3 Å (right). **b)** Cryo-EM map of OCT4 region from the OCT4-nucleosome complex. Focused classification and refinements improved the resolution of this 30 kDa fragment to 4.2 Å. **c)** Angular distribution for OCT4. **d)** The fourier shell correlation (FSC) curve showing the resolution of the map. **e)** Directional FSC plot showing uniform resolution in all directions. **f)** The OCT4-DNA density from b) was modified in Phenix, which improved the resolution to 3.9 Å. The map is colored by local resolution. Fourier shell correlation (FSC) curve showing the resolution

is shown on the right. **g)** The model of the OCT4 bound to DNA (PDB: 3L1P) was refined into the cryo-EM map. The representative region showing map quality and fit of the model is shown on the right. Red line shows the kink in the DNA. **h)** A composite cryo-EM map of OCT4 bound to the *LIN28B* nucleosome containing 182bp of DNA at 2.8-3.9 Å resolution (left). Model for the cryo-EM structure is shown on the right. **i)** DNA sequence and schematic representation showing *LIN28B* DNA positioning on the OCT4-nucleosome complex. OCT4, SOX2, KLF4 and c-MYC binding sites are labeled. The cleavage site for the restriction enzyme Mnl I (Fig. 3c) is marked with an arrow.

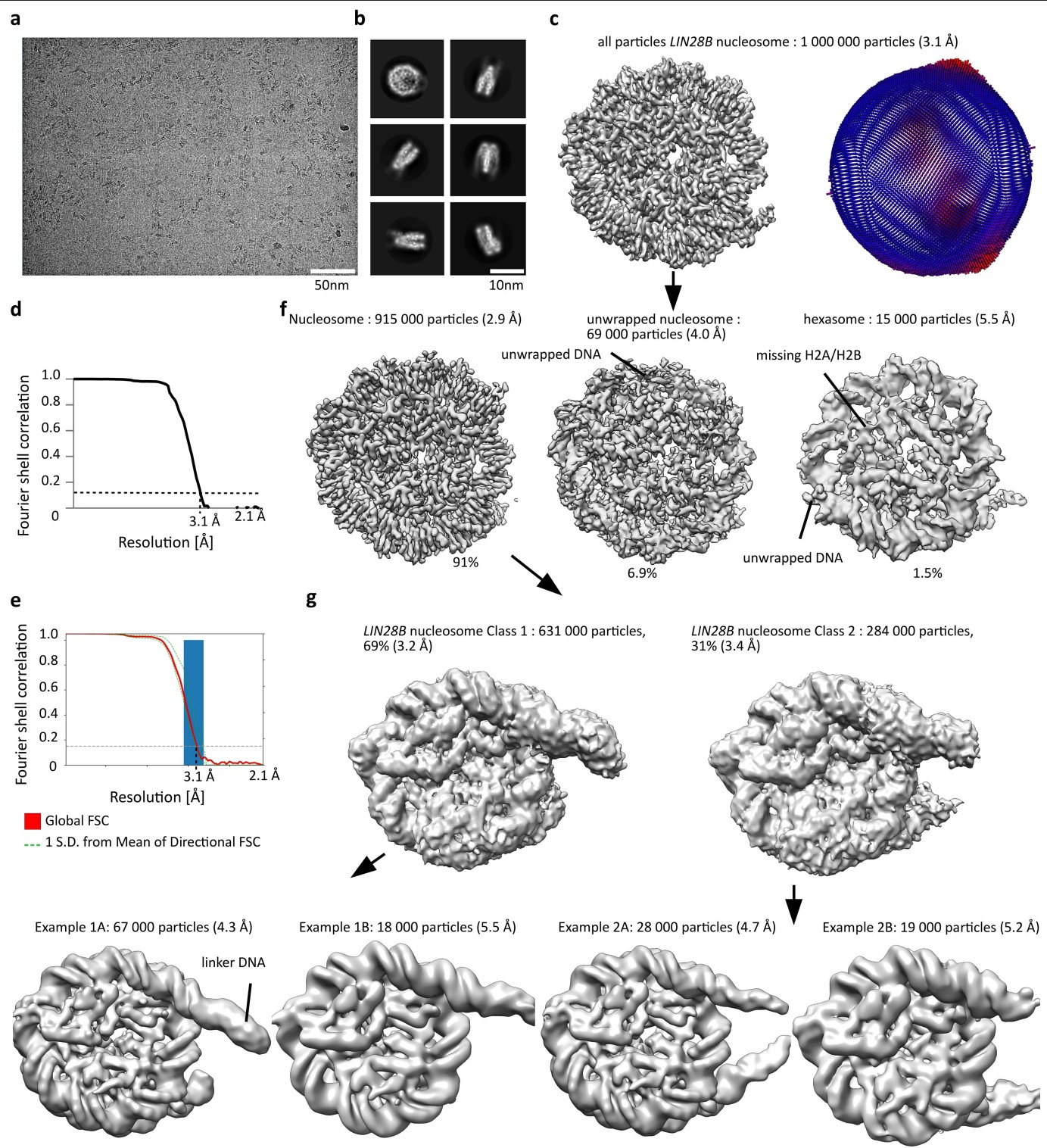

**a**

**b**
50nm
10nm

**c**
all particles *LIN28B* nucleosome : 1 000 000 particles (3.1 Å)

**d**

Fourier shell correlation
Resolution [Å]
3.1 Å
2.1 Å

**e**

Fourier shell correlation
Resolution [Å]
3.1 Å
2.1 Å
— Global FSC
--- 1 S.D. from Mean of Directional FSC

**f**
Nucleosome : 915 000 particles (2.9 Å)

unwrapped nucleosome :
69 000 particles (4.0 Å)
unwrapped DNA

hexasome : 15 000 particles (5.5 Å)
missing H2A/H2B
unwrapped DNA

91%
6.9%
1.5%

**g**
*LIN28B* nucleosome Class 1 : 631 000 particles, 69% (3.2 Å)

*LIN28B* nucleosome Class 2 : 284 000 particles, 31% (3.4 Å)

Example 1A: 67 000 particles (4.3 Å)
linker DNA

Example 1B: 18 000 particles (5.5 Å)

Example 2A: 28 000 particles (4.7 Å)

Example 2B: 19 000 particles (5.2 Å)

**Extended Data Fig. 3 | Cryo-EM of *LIN28B* nucleosome. a)** Representative cryo-EM micrograph from a set of 6000 micrographs collected with Titan Krios electron microscope at 300 keV. Nucleosome particles in multiple orientations are visible. **b)** Representative 2D class averages showing nucleosomes. Many details in nucleosomes are visible in 2D class averages. **c)** Cryo-EM map of nucleosome from the entire dataset, refined to 3.1 Å. Angular distribution for nucleosome is shown on the right. **d)** Fourier shell correlation (FSC) curve showing the resolution of the map in c). **e)** Directional FSC plot showing uniform resolution in all directions. **f)** Classification of the *LIN28B* nucleosome dataset resulted in three classes of nucleosome like particles (nucleosome, unwrapped nucleosome and hexasome). Number of particles corresponding to each class is indicated. **g)** Classification of *LIN28B* nucleosome particles from f) showing that DNA protrudes on both sides of nucleosome in some classes.

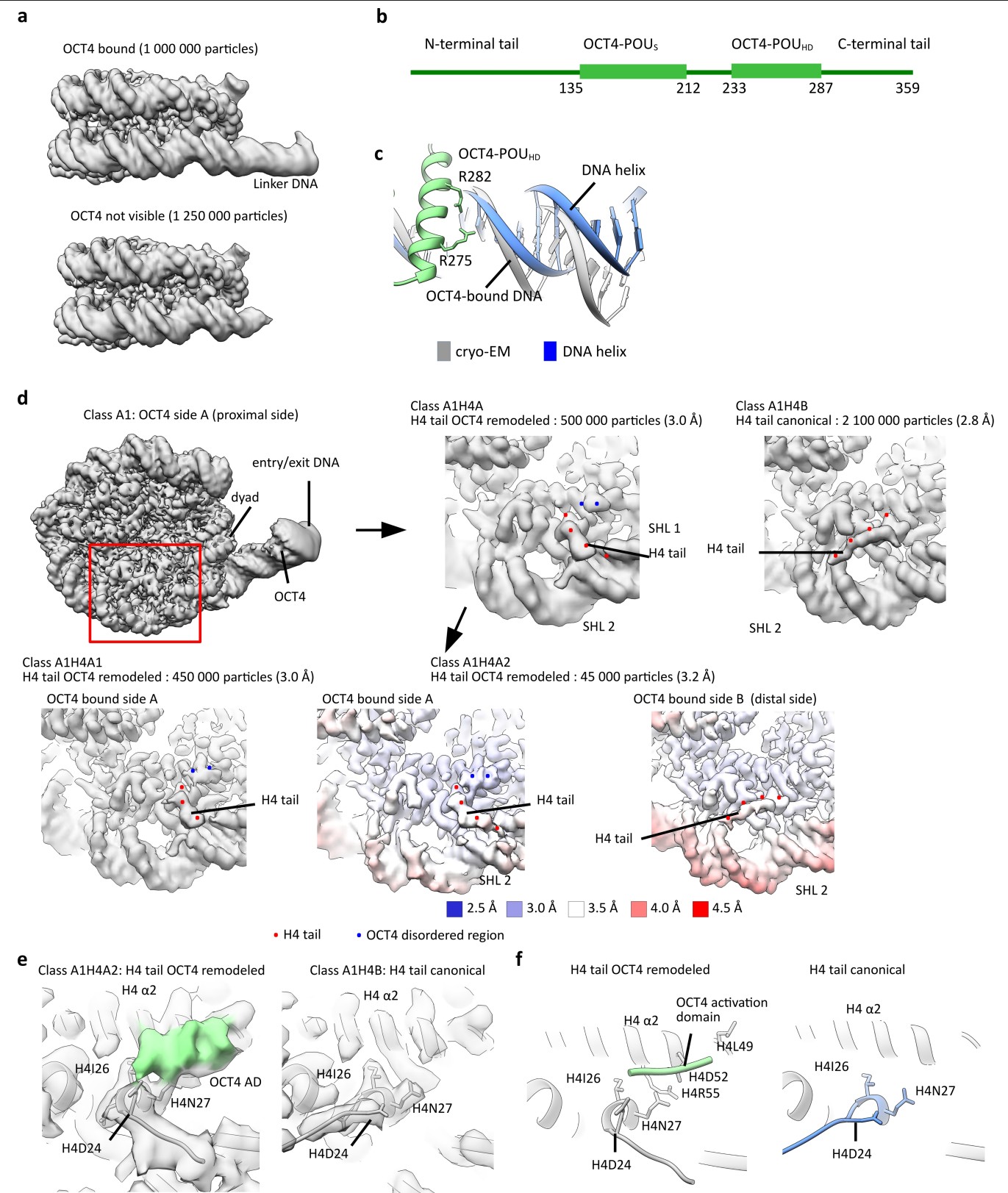

**a**

OCT4 bound (1 000 000 particles)

Linker DNA

OCT4 not visible (1 250 000 particles)

**b**

N-terminal tail | OCT4-POU$_S$ | OCT4-POU$_{HD}$ | C-terminal tail

135 212 233 287 359

**c**

OCT4-POU$_{HD}$
R282
DNA helix
R275
OCT4-bound DNA

cryo-EM DNA helix

**d**

Class A1: OCT4 side A (proximal side)

entry/exit DNA
dyad
OCT4

Class A1H4A
H4 tail OCT4 remodeled : 500 000 particles (3.0 Å)

SHL 1
H4 tail
SHL 2

Class A1H4B
H4 tail canonical : 2 100 000 particles (2.8 Å)

H4 tail
SHL 2

Class A1H4A1
H4 tail OCT4 remodeled : 450 000 particles (3.0 Å)

OCT4 bound side A
H4 tail

Class A1H4A2
H4 tail OCT4 remodeled : 45 000 particles (3.2 Å)

OCT4 bound side A
H4 tail
SHL 2

OCT4 bound side B (distal side)
H4 tail
SHL 2

2.5 Å 3.0 Å 3.5 Å 4.0 Å 4.5 Å

• H4 tail • OCT4 disordered region

**e**

Class A1H4A2: H4 tail OCT4 remodeled

H4 α2
OCT4 AD
H4I26
H4N27
H4D24

Class A1H4B: H4 tail canonical

H4 α2
H4I26
H4N27
H4D24

**f**

H4 tail OCT4 remodeled

H4 α2
OCT4 activation domain
H4L49
H4D52
H4R55
H4I26
H4N27
H4D24

H4 tail canonical

H4 α2
H4I26
H4N27
H4D24

**Extended Data Fig. 4** | See next page for caption.

**Extended Data Fig. 4 | Intrinsically disordered region of OCT4 binds to histone H4. a)** Cryo-EM maps of OCT4_nucleosome complex and *LIN*28B nucleosome. DNA extends only on OCT4 bound side of the nucleosome in the OCT4 bound sample. Linker DNA is more defined and stabilized by OCT4. **b)** Schematic representation of OCT4 domains. The POU$_S$ and POU$_{HD}$ domains are structured, whereas the N- and C-terminal tails are disordered. **c)** Model of OCT4 bound to the linker DNA showing the kink in the DNA introduced by binding of OCT4-POU$_{HD}$. Arg in OCT4-POU$_{HD}$ interact with DNA. **d)** Classification of the cryo-EM data revealed two conformations of the H4 tail on the OCT4 proximal side. Red dots depict the H4 tail. The class with the re-positioned H4 tail, which goes to the SHL1, contains an additional density which is labeled with blue dots. Maps are colored by local resolution. **e)** Cryo-EM maps and fitted models showing two positions of the H4 tail. Note different orientation of H4D24 which interacts with the OCT4 density. OCT4 density is shown in green. **f)** Detailed views of the H4 α2 and N-terminal tail on the OCT4-proximal side of the nucleosome, showing the two distinct conformations of the H4 tail, canonical (blue, left) and OCT4 remodeled (grey, right). The disordered region of OCT4 that interacts with H4 is represented in green, and it contacts Asp24 and Asn27 in H4 tail and Asp52 and Arg55 in H4 α2.

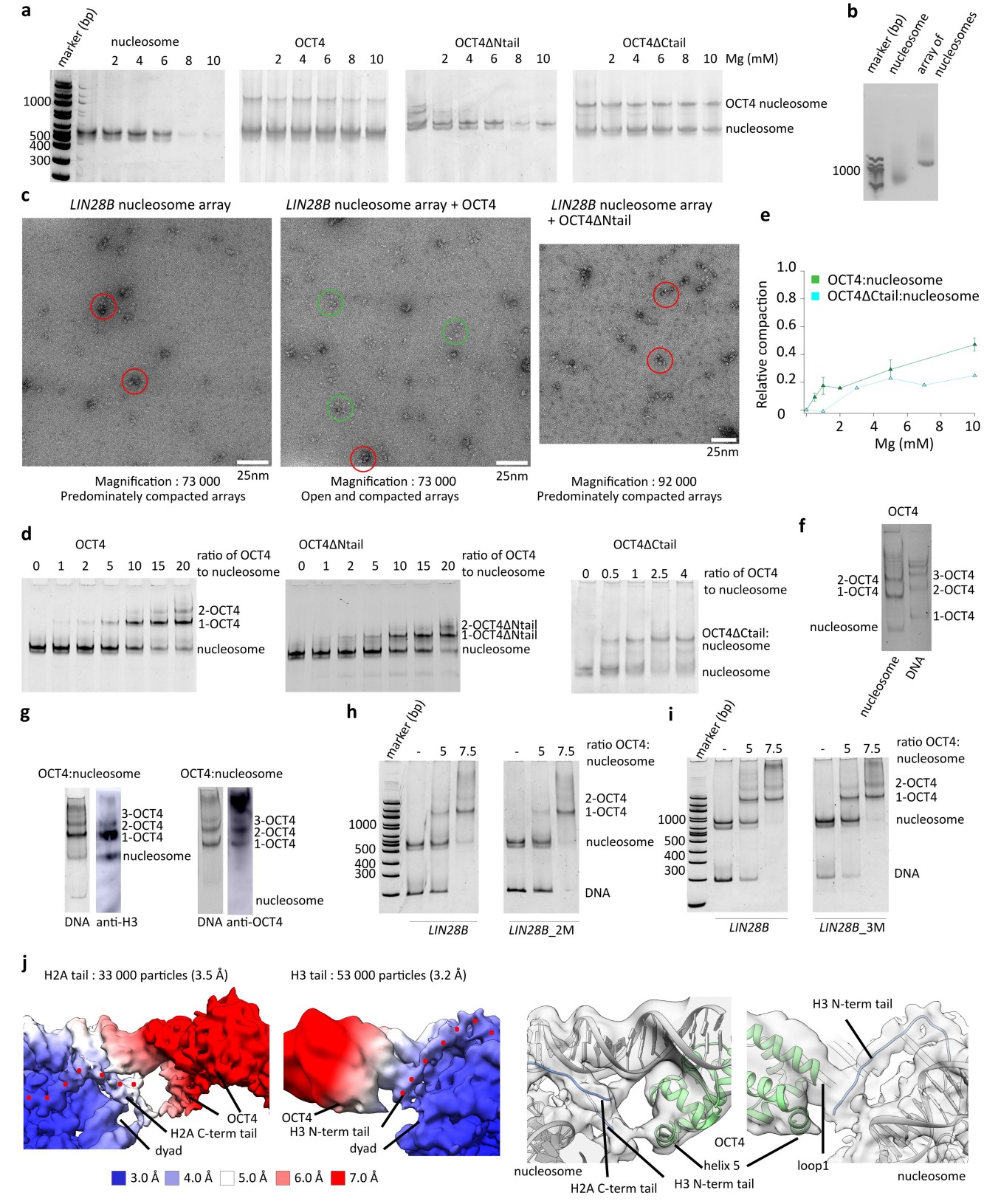

**Extended Data Fig. 5** | See next page for caption.

**Extended Data Fig. 5 | N-terminal disordered region of OCT4 is required for chromatin de-compaction. a)** Native gels showing $Mg^{2+}$ induced compaction of nucleosomes and nucleosomes bound to OCT4 (n = 4), OCT4ΔNtail (n = 4) and OCT4ΔCtail (n = 3). OCT4 binding reduces nucleosome compaction. Deletion of OCT4 N-terminal disordered region eliminates OCT4 effect on nucleosome compaction. **b)** Native agarose gel showing assembly of nucleosome and nucleosome array (n > 3). **c)** Negative stain micrographs showing $Mg^{2+}$ induced compaction of the *LIN28B* nucleosome array (n = 26 micrographs), the OCT4 bound *LIN28B* nucleosome array (n=32 micrographs) and the OCT4ΔNtail bound *LIN28B* nucleosome array (n = 23 micrographs). Most nucleosomes are compacted (red circle) in sample containing *LIN28B* arrays and OCT4ΔNtail bound *LIN28B* arrays. Many more open arrays (green circle) are detectable when OCT4 is bound to *LIN28B* arrays. **d)** Native gels showing binding of OCT4ΔNtail (n = 4) and OCT4ΔCtail (n = 2) to the *LIN28B* nucleosome. OCT4ΔNtail and OCT4ΔCtail binds nucleosome comparably to wild type OCT4. **e)** Quantification of data from d). Deletion of OCT4 C-terminal disordered tail does not reduce Oct4 effect on nucleosome compaction. Data are mean and s.e.m., n = 4. **f)** A native gel stained for DNA showing OCT4 binding to nucleosome and DNA. Binding to DNA and nucleosome generates distinct bands. **g)** A native gel stained for DNA showing OCT4 binding to nucleosome and western blot with anti-H3 showing presence of histones in these complexes (left). A native gel stained for DNA showing OCT4 binding to nucleosome and western blot with anti-OCT4 showing presence of OCT4 in these complexes (right). Each experiment has been performed > 3 times. **h)** Native gel showing OCT4 binding to the *LIN28B* nucleosome and *LIN28B* nucleosome with mutated binding site 2 (*LIN28B*-2M). Mutation of the binding site 2 did not affect binding of 1st OCT4. **i)** Native gel showing OCT4 binding to the *LIN28B* nucleosome and *LIN28B* nucleosome with mutated binding site 3 (*LIN28B*-3M). Mutation of the binding site 3 did not affect binding of 1st OCT4. **j)** Cryo-EM maps from a subset of data showing OCT4 interaction with H3 and H2A tails. The maps are colored by local resolution. Histone tails are marked with red dots. Model showing interaction of OCT4 with histone tails is shown below. Model of OCT4-nucleosome complex was rigid body fitted into cryo-EM maps.

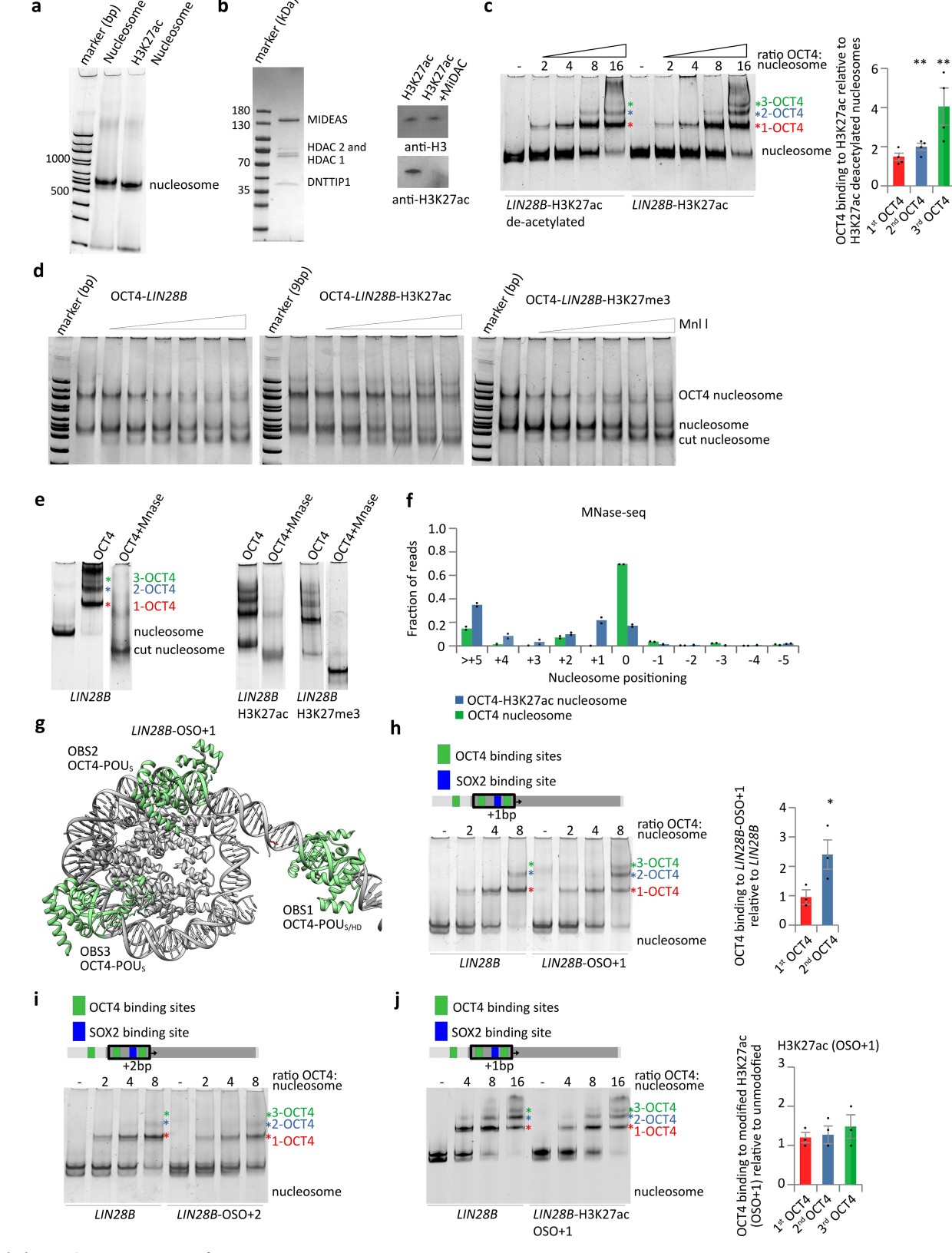

**Extended Data Fig. 6** | See next page for caption.

**Extended Data Fig. 6 | Histone modifications modulate OCT4 cooperativity.**
**a)** Representative native gel from 5 independent experiments showing assembly of unmodified and H3K27ac *LIN28B* nucleosomes. **b)** Left, a representative SDS-PAGE from 3 independent gels showing purification of MiDAC complex. Right, western blot with anti-H3 and anti-H3K27ac showing deacetylation of H3K27ac. **c)** Left, representative native gel electrophoresis showing OCT4 binding to the deacetylated *LIN28B*-H3K27ac or *LIN28B*-H3K27ac nucleosomes. The composition of the OCT4-bound bands was validated by Western blot (Extended Data Fig. 5g). Colored asterisks indicate the number of OCT4 molecules bound to the nucleosome: red, 1 OCT4; blue, 2 OCT4; and green, 3 OCT4. Right, quantification of OCT4 binding to *LIN28B*-H3K27ac nucleosome relative to the deacetylated *LIN28B*-H3K27ac; data are mean and s.e.m. of 4 independent experiments; ** p = 0.001 and p = 0.005 for 2$^{nd}$ and 3$^{rd}$ OCT4, one-sided Student's t-test. **d)** Representative gel from 4 independent experiments of Mnl I digestion of unmodifed and H3K27ac nucleosomes bound to OCT4. Binding of OCT4 to nucleosomes increases Mnl I digestion of nucleosome indicating exposure of Mnl I site. OCT4 bound to H3K27ac nucleosomes shows decreased degradation at Mnl I site compared to unmodified nucleosomes bound to OCT4. **e)** Native gel showing OCT4 binding and MNase digestion of OCT4 bound unmodified and H3K27ac *LIN28B* nucleosomes. **f)** Quantification of sequencing of Mnase I digested OCT4-bound nucleosomes (unmodified and H3K27ac). The y-axis shows fraction of nucleosome size reads starting at defined position, the x-axis shows position of the first base pair relative to the most abundant position (0 as observed in the structure). Data are mean and spread of 2 independent experiments. **g)** Model of OCT4 bound to the *LIN28B* nucleosome with OBS2/3 and SOX2 binding sites moved for +1bp (OSO+1), showing OCT4-binding sites on DNA in green. OCT4 bound to OBS1 is in solid green; OCT4 structure was superimposed on OBS2 and on OBS3. In this conformation, OCT4-POU$_S$ can bind to OBS2 and OBS3. Note, shift of 1bp exposes OCT4-POU$_S$ binding site at OBS2 instead of OCT4-POU$_{HD.}$ **h)** Left, representative native gel electrophoresis showing OCT4 binding to the *LIN28B* or *LIN28B* nucleosomes with OBS2/3 and SOX2 binding sites moved for +1bp (OSO+1). Right, quantification of the native gel electrophoresis, data shown as s.e.m. of 3 independent experiments. Bands marked with * were used for quantification. * p = 0.02, one-sided Student's t-test. **i)** Representative native gel electrophoresis from 2 independent experiments showing OCT4 binding to the *LIN28B* or *LIN28B* nucleosomes with OBS2/3 and SOX2 binding sites moved for +2bp (OSO+2). **j)** Left, representative native gel electrophoresis showing OCT4 binding to the *LIN28B* or H3K27ac *LIN28B* nucleosomes with OBS2/3 and SOX2 binding sites moved for +1bp (OSO+1). Right, quantification of the native gel electrophporesis, data shown as s.e.m. of 3 independent experiments. Bands marked with * were used for quantification.

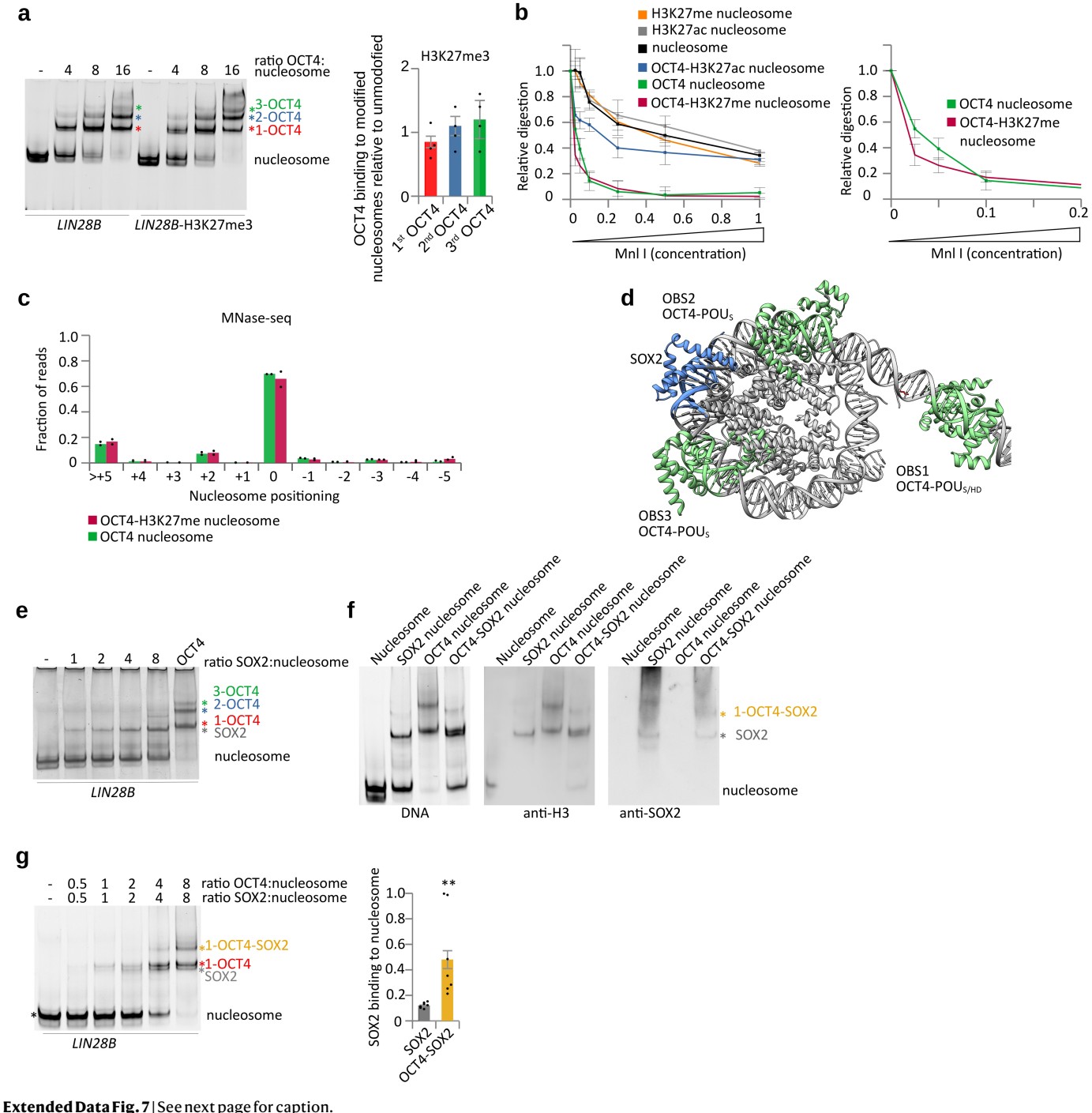

**Extended Data Fig. 7** | See next page for caption.

**Extended Data Fig. 7 | Histone modifications modulate OCT4 and SOX2 cooperativity. a)** Left, representative native gel electrophoresis showing OCT4 binding to the *LIN28B* or H3K27me3 *LIN28B* nucleosomes. Colored asterisks indicate the number of OCT4 molecules bound to the nucleosome: red, 1 OCT4; blue, 2 OCT4; and green, 3 OCT4. Right, quantification of OCT4 binding to H3K27me3 *LIN28B* relative to unmodified *LIN28B*, data shown as s.e.m. of 4 independent experiments. Bands marked with * were used for quantification. **b)** Left, quantification of MnlI digestion of free and OCT4-bound nucleosomes (unmodified, H3K27ac and H3K27me3). The y-axis shows intensity of nucleosome bands after enzyme digestion, normalized to the input (without enzyme). Data are mean and s.e.m. of 4 independent experiments. Representative gels are shown in Extended Data Fig. 6d. Right, comparison of MnlI digestion between unmodified and H3K27me3 OCT4 bound nucleosomes. **c)** Quantification of sequencing of Mnase I digested OCT4-bound nucleosomes (unmodified or H3K27me3). The y-axis shows fraction of nucleosome size reads starting at defined position, the x-axis shows position of the first base pair relative to the most abundant position (0 as observed in the structure). Data are mean and spread of 2 independent experiments. **d)** Model of OCT4 bound to the *LIN28B* nucleosome with SOX2 binding site shown in blue. OCT4 bound to OBS1 is in solid green; OCT4 structure was superimposed on OBS2

and on OBS3. SOX2 binding site and SOX2 model are shown in blue. **e)** A representative native gel from >6 independent experiments stained for DNA showing SOX2 and OCT4 binding to nucleosome. **f)** Left, native gel stained for DNA showing SOX2 and OCT4 binding to nucleosome. Center, western blot with anti-H3 showing presence of histones in the complexes. Right, western blot with anti-Sox2 showing presence of Sox2 in these complexes. Each experiment has been replicated >3 times. **g)** Left, representative native gel electrophoresis showing OCT4 and SOX2 binding to the *LIN28B* nucleosome. OCT4 and SOX2 were mixed and added to nucleosomes as indicated. SOX2 binding to nucleosome was validated by western blot analyses (Extended Data Fig. 7f). Colored asterisks indicate the molecules bound to the nucleosome: black, nucleosome alone; red, 1 OCT4; gray, 1 SOX2; orange, 1 OCT4 and 1 SOX2. Right, quantification of SOX2 binding, with data shown as mean and s.e.m. of 5 independent experiments; ** p = 0.003, one-sided Student's t-test. To assess SOX2 binding to OCT4-bound nucleosome, we used 1-OCT4-SOX2 (orange asterisk) and 1-OCT4 (red asterisk) bands; to assess SOX2 binding to nucleosome, we used SOX2-bound nucleosome (gray asterisk) and input nucleosome (black asterisk) bands. See Methods and quantification data in Supplementary Table 3.

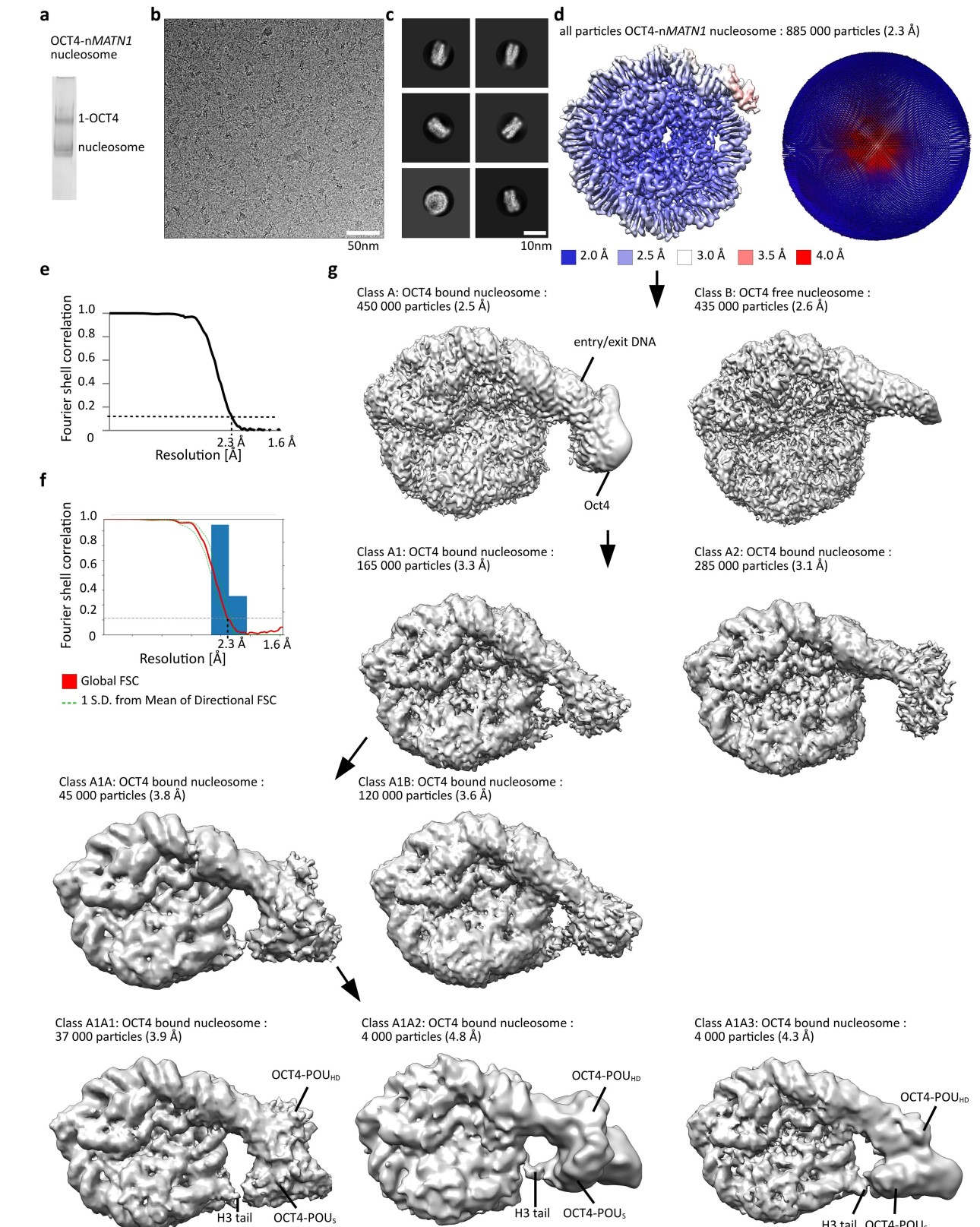

**Extended Data Fig. 8 | Cryo-EM of OCT4 bound to *nMATN1* nucleosome.**
**a)** Representative native PAGE from >5 independent experiments showing OCT4 binding to *nMATN1* nucleosome. **b)** Representative cryo-EM micrograph from 35 000 images collected with Titan Krios electron microscope at 300 keV. Nucleosome particles in multiple orientations are visible. **c)** Representative 2D class averages showing nucleosomes. Many details in nucleosomes are visible in 2D class averages. **d)** Cryo-EM map of nucleosome from the entire dataset, refined to 2.3 Å. The map is colored by local resolution. Angular distribution for nucleosome is shown on the right. **e)** Fourier shell correlation (FSC) curve showing the resolution of the map in d). **f)** Directional FSC plot showing uniform resolution in all directions. **g)** Classification of the nucleosome from d). Classification revealed two classes, nucleosome and nucleosome with bound OCT4. OCT4 bound class was further classified revealing many conformations of OCT4 bound to linker DNA. Two conformations with OCT4 bound to linker DNA are shown. Note OCT4 interaction with the H3 tail.

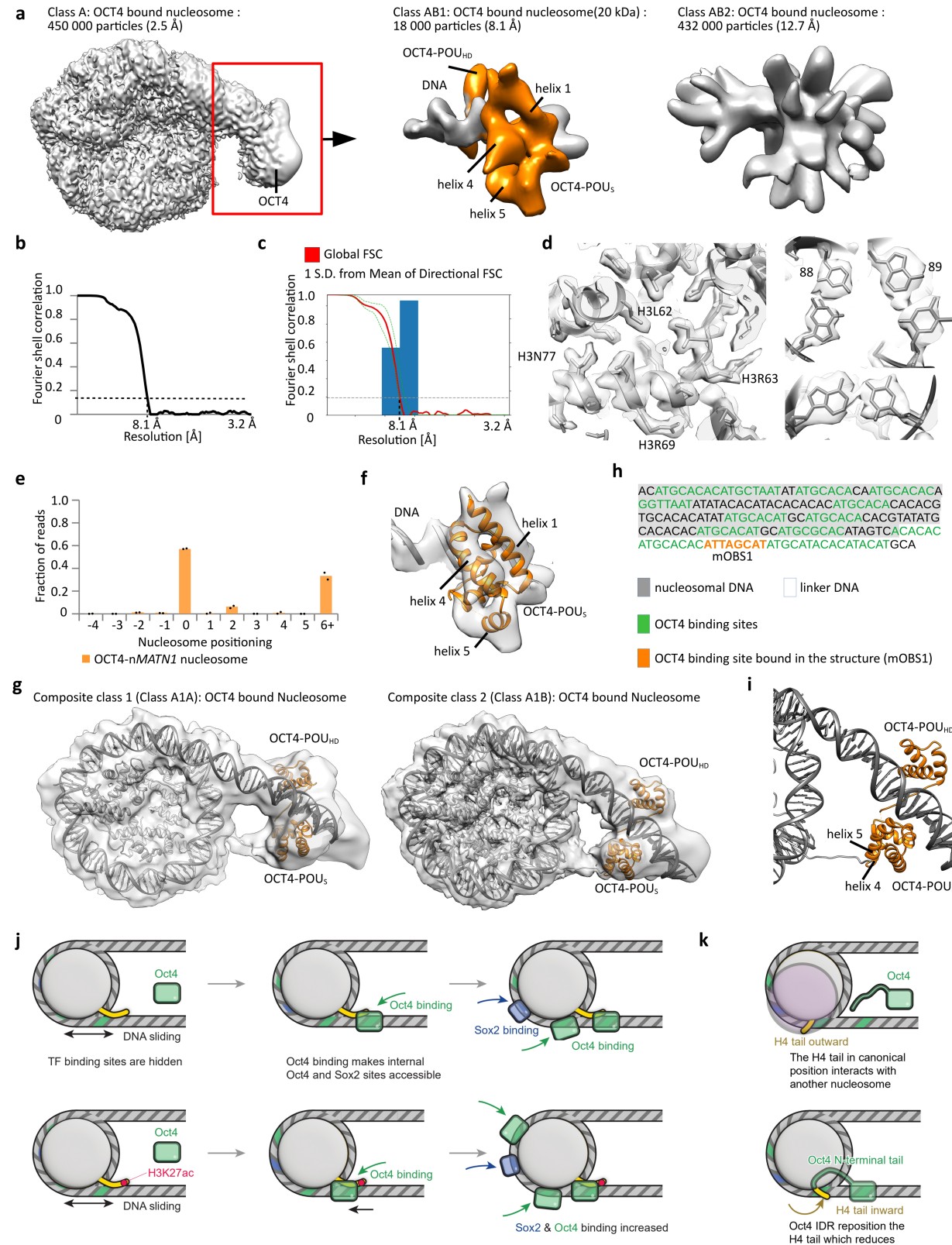

**a** Class A: OCT4 bound nucleosome : 450 000 particles (2.5 Å)

Class AB1: OCT4 bound nucleosome(20 kDa) : 18 000 particles (8.1 Å)

OCT4-POU_HD
DNA
helix 1
helix 4
OCT4-POU_S
helix 5

OCT4

Class AB2: OCT4 bound nucleosome : 432 000 particles (12.7 Å)

**b**
Fourier shell correlation
Resolution [Å]
8.1 Å   3.2 Å

**c**
Global FSC
1 S.D. from Mean of Directional FSC
Fourier shell correlation
Resolution [Å]
8.1 Å   3.2 Å

**d**
H3L62
H3N77
H3R63
H3R69
88   89

**e**
Fraction of reads
Nucleosome positioning
-4 -3 -2 -1 0 1 2 3 3 4 5 6+
OCT4-n*MATN1* nucleosome

**f**
DNA
helix 1
helix 4
OCT4-POU_S
helix 5

**h**
ACATGCACACATGCTAATATATGCACACAATGCACACA
GGTTAATATATACACATACACACACATGCACACACACG
TGCACACATATATGCACATGCATGCACACACGTATATG
CACACACATGCACATGCGATGCGCACATAGTCACACAC
ATGCACACATTAGCATATGCATACACATACATGCA
mOBS1

■ nucleosomal DNA   □ linker DNA
■ OCT4 binding sites
■ OCT4 binding site bound in the structure (mOBS1)

**g** Composite class 1 (Class A1A): OCT4 bound Nucleosome

OCT4-POU_HD
OCT4-POU_S

Composite class 2 (Class A1B): OCT4 bound Nucleosome

OCT4-POU_HD
OCT4-POU_S

**i**
OCT4-POU_HD
helix 5
helix 4
OCT4-POU_S

**j**
Oct4
DNA sliding
TF binding sites are hidden

Oct4 binding
Oct4 binding makes internal Oct4 and Sox2 sites accessible

Sox2 binding
Oct4 binding

Oct4
H3K27ac
DNA sliding

Oct4 binding

Sox2 & Oct4 binding increased

**k**
Oct4
H4 tail outward
The H4 tail in canonical position interacts with another nucleosome

Oct4 N-terminal tail
H4 tail inward
Oct4 IDR reposition the H4 tail which reduces chromatin interaction

**Extended Data Fig. 9** | See next page for caption.

**Extended Data Fig. 9 | Histone modifications modulate OCT4 and SOX2 cooperativity on various human DNA. a)** Cryo-EM map of OCT4 region from the OCT4_nucleosome complex from Fig. 8g). Focused classification and refinements improved the resolution of this 20 kDa fragment to 8.1 Å. **b)** Fourier shell correlation (FSC) curve showing the resolution of the map in a). **c)** Directional FSC plot showing uniform resolution in all directions. **d)** Representative regions showing map quality and fit of the model are shown for the nucleosome with bound OCT4. Right: bases in the DNA are well resolved. **e)** Quantification of sequencing of Mnase I digested OCT4-bound *nMATN1* nucleosomes. The y-axis shows fraction of nucleosome size reads starting at defined position, the x-axis shows position of the first base pair relative to the most abundant position (0 as observed in the structure). Data are mean and spread of 2 independent experiments. **f)** The model of the OCT4 bound to DNA (Extended Data Fig. 2g) was refined into the cryo-EM map. The representative region showing map quality and fit of the model is shown. **g)** Cryo-EM models of OCT4 bound to the *nMATN1* nucleosome containing 186bp of DNA at 2.2-5.6 Å resolution for two most dominant conformations. **h)** DNA sequence and schematic representation showing *nMATN1* DNA positioning on the OCT4_nucleosome complex. Potential OCT4 binding sites are labeled in green.

OCT4 binding site occupied in the structure is labeled in orange. **i)** Close-up views of the nucleosome entry/exit site showing interaction of the OCT4_POU$_S$ domain with the H3 N-terminal tail. Ribbon representation shows OCT4_POU$_S$ helix 4 and helix 5 interacting with histone H3 N-terminal tail. **j)** In *LIN28B* nucleosome OCT4 (light green) and SOX2 (light blue) binding sites are wrapped around the histone octamer. *LIN28B* nucleosomes are mobile, and nucleosome sliding transiently exposes the OCT4 binding site 1 (green), which leads to binding of OCT4 (green box). OCT4 binding (green box) traps DNA in a more defined position, which exposes internal OCT4 and SOX2 binding sites (blue). OCT4 bound to the OBS1 interacts with the histone H3 tail. H3K27ac modifies this interaction leading to DNA movement towards the histone octamer, which exposes internal OCT4 and SOX2 sites even more, leading to increased binding. **k)** The canonical H4 tail conformation (yellow, facing outward) favors inter-nucleosome interactions by interacting with the acidic patch of neighboring nucleosomes. These interactions are essential for chromatin compaction. OCT4 DNA binding domain binds linker DNA whereas disordered activation domain binds H4 near the H4 tail. This repositions the H4 tail to an inward conformation that reduces inter-nucleosome interactions in chromatin.

**Extended Data Table 1 | Cryo-EM data collection, refinement, and validation statistics**

| | OCT4 bound *LIN28B* nucleosome EMD-29850 PDB ID 8G8G | OCT4/linker *LIN28B* DNA EMD- 29846 PDB ID 8G8E | OCT4 bound *nMATN1* nucleosome EMD-29843 PDB ID 8G88 | OCT4/linker *nMATN1* DNA EMD-29841 PDB ID 8G87 |
|---|---|---|---|---|
| **Data collection and processing** | | | | |
| Magnification | | | | |
| Voltage (kV) | 300 | 300 | 300 | 300 |
| Electron exposure (e–/Å$^2$) | 90 | 90 | 50 | 50 |
| Defocus range (μm) | -0.7 – -2.5 | -0.7– -2.5 | -0.7 – -2.5 | -0.7 – -2.5 |
| Pixel size (Å) | 1.06 | 1.06 | 0.805 | 0.805 |
| Symmetry imposed | C1 | C1 | C1 | C1 |
| Initial particle images (no.) | ~ 2 620 000 | ~ 2 620 000 | ~ 850 000 | ~ 850 000 |
| Final particle images (no.) | ~ 1 000 000 | ~ 65 000 | ~ 450 000 | ~ 18 000 |
| Map resolution (Å) FSC threshold | 2.8 | 3.9 | 2.3 | 8.1 |
| Map resolution range (Å) | 2.5-3.5 | 3.5-5.0 | 2.2-3.5 | 8-15 |
| | | | | |
| **Refinement** | | | | |
| Initial model used | 6WZ5 | | | |
| Model resolution (Å) FSC threshold | 2.8 | 3.9 | 2.2 | 8.1 |
| Model resolution range (Å) | 235-2.6 | 235-3.6 | 235-2.2 | 235-6 |
| Model composition Nonhydrogen atoms Protein residues Nucleotide | 14667 935 352 | 2484 143 64 | 14183 904 347 | |
| *B* factors (Å$^2$) Protein Nucleotide | 335.97 377.41 | 225.42 243.60 | 250.90 268.11 | |
| R.m.s. deviations Bond lengths (Å) Bond angles (°) | 0.01 0.987 | 0.007 0.935 | 0.019 1.5 | |
| Validation MolProbity score Clashscore Poor rotamers (%) | 1.89 23.96 0.13 | 2.05 25.42 0 | 2.04 35.74 0 | |
| Ramachandran plot Favored (%) Allowed (%) | 97.92 2.08 | 97.12 2.88 | 98.08 1.92 | |

# Reporting Summary

## Statistics

For all statistical analyses, confirm that the following items are present in the figure legend, table legend, main text, or Methods section.

| n/a | Confirmed | |
|---|---|---|
| ☐ | ☒ | The exact sample size (*n*) for each experimental group/condition, given as a discrete number and unit of measurement |
| ☒ | ☐ | A statement on whether measurements were taken from distinct samples or whether the same sample was measured repeatedly |
| ☐ | ☒ | The statistical test(s) used AND whether they are one- or two-sided<br>*Only common tests should be described solely by name; describe more complex techniques in the Methods section.* |
| ☒ | ☐ | A description of all covariates tested |
| ☒ | ☐ | A description of any assumptions or corrections, such as tests of normality and adjustment for multiple comparisons |
| ☐ | ☒ | A full description of the statistical parameters including central tendency (e.g. means) or other basic estimates (e.g. regression coefficient) AND variation (e.g. standard deviation) or associated estimates of uncertainty (e.g. confidence intervals) |
| ☐ | ☒ | For null hypothesis testing, the test statistic (e.g. *F*, *t*, *r*) with confidence intervals, effect sizes, degrees of freedom and *P* value noted<br>*Give P values as exact values whenever suitable.* |
| ☒ | ☐ | For Bayesian analysis, information on the choice of priors and Markov chain Monte Carlo settings |
| ☒ | ☐ | For hierarchical and complex designs, identification of the appropriate level for tests and full reporting of outcomes |
| ☒ | ☐ | Estimates of effect sizes (e.g. Cohen's *d*, Pearson's *r*), indicating how they were calculated |

*Our web collection on statistics for biologists contains articles on many of the points above.*

## Software and code

Policy information about availability of computer code

| Data collection | Serial EM for data collection |
|---|---|
| Data analysis | Relion, CtfFind, Phenix, MotionCor, Coot |

For manuscripts utilizing custom algorithms or software that are central to the research but not yet described in published literature, software must be made available to editors and reviewers. We strongly encourage code deposition in a community repository (e.g. GitHub). See the Nature Portfolio guidelines for submitting code & software for further information.

## Data

Policy information about availability of data

All manuscripts must include a data availability statement. This statement should provide the following information, where applicable:
- Accession codes, unique identifiers, or web links for publicly available datasets
- A description of any restrictions on data availability
- For clinical datasets or third party data, please ensure that the statement adheres to our policy

EM densities and models have been deposited in the Electron Microscopy Data Bank and PDB under accession codes P DB 8G8G for Oct4 bound to Lin28B nucleosome built using maps EMD-29855 (Oct4 _Nucleosome, all particles), EMD-29850 (Oct4 _Nucleosome, H3 tail subset), E M D-29852 (Oct4 _Nucleosome, H2A tail subset), EMD-29853 (Oct4 _Nucleosome, H4 tail A subset), EMD-29854 (Oct4 _Nucleosome, H4 tail B subset), EMD-29846 (Oct4 _Nucleosome, Oct4 focus) and

PDB 8G8E. For Oct4 bound to n Matnl nucleosome following maps and coordinates were deposited: EMD-29837 and PDB 8G86 (Oct4 _Nucleosome, nucleosome focus); EMD-29841 and PDB 8G87 (Oct4 _Nucleosome, Oct4 focus); EMD-29843 and PDB 8G88 (Oct4_Nucleosome, conformation 1); EMD-29845 and PDB 8G8B (Oct4 _Nucleosome, conformation 2).

## Human research participants

Policy information about studies involving human research participants and Sex and Gender in Research.

| | |
|---|---|
| Reporting on sex and gender | NA |
| Population characteristics | NA |
| Recruitment | NA |
| Ethics oversight | NA |

Note that full information on the approval of the study protocol must also be provided in the manuscript.

# Field-specific reporting

Please select the one below that is the best fit for your research. If you are not sure, read the appropriate sections before making your selection.

☒ Life sciences          ☐ Behavioural & social sciences          ☐ Ecological, evolutionary & environmental sciences

For a reference copy of the document with all sections, see nature.com/documents/nr-reporting-summary-flat.pdf

# Life sciences study design

All studies must disclose on these points even when the disclosure is negative.

| | |
|---|---|
| Sample size | This has been described in the corresponding figure legends. No sample size calculation was performed. For the biochemical experiments, we have performed independent experiments, which is sufficient to establish the variation. |
| Data exclusions | In the EM analysis, only junk particles have been removed using 2D and 3D classifications in RELION. No other data exclusions involved in analysis. |
| Replication | Biochemical assays have been replicated and the number of replicates has been mentioned in the figure legends. for each experiment. The cryo-EM structure determination generally does not involve repeat of the experiments, specially because of the final structure is the result of ensemble averaging of several thousand particles. |
| Randomization | Because there was no sub-group analysis involved and the sample size was small, no randomization was done. |
| Blinding | Blinding was not applicable to this study. Biochemical experiments were visualized using fluorescence and quantification of the bands was done using software (Quantity One, Biorad), requiring no subjective analysis. Likewise, cryoEM data was imaged and processed using standard software packages requiring no subjective judgement. |

# Reporting for specific materials, systems and methods

We require information from authors about some types of materials, experimental systems and methods used in many studies. Here, indicate whether each material, system or method listed is relevant to your study. If you are not sure if a list item applies to your research, read the appropriate section before selecting a response.

## Materials & experimental systems

| n/a | Involved in the study |
|---|---|
| ☐ | ☒ Antibodies |
| ☐ | ☒ Eukaryotic cell lines |
| ☐ | ☐ Palaeontology and archaeology |
| ☐ | ☐ Animals and other organisms |
| ☐ | ☐ Clinical data |
| ☐ | ☐ Dual use research of concern |

## Methods

| n/a | Involved in the study |
|---|---|
| ☐ | ☐ ChIP-seq |
| ☐ | ☐ Flow cytometry |
| ☐ | ☐ MRI-based neuroimaging |

# Antibodies

| | |
|---|---|
| Antibodies used | anti-Oct4 antibody (Abcam ab109183), HRP-conjugated anti-His antibody (Invitrogen – Thermo Fisher R931-25), anti-H3 antibody (abcam ab1791), anti-Sox2 antibody (Abcam ab 92494), HRP-conjugated anti-rabbit secondary antibody (Biorad 170-6515) |
| Validation | anti-Oct4 antibody (abcam ab109183):<br>validation for Oct4 band in NCCIT (Human pluripotent embryonic carcinoma epithelial cell) whole cell lysate<br>(https://www.abcam.com/oct4-antibody-epr2054-ab109183.html)<br><br>HRP-conjugated anti-His antibody (Invitrogen – Thermo Fisher, Catalog # R931-25):<br>https://www.thermofisher.com/antibody/product/6x-His-Tag-Antibody-clone-3D5-Monoclonal/R931-25<br><br>anti-H3 antibody (abcam ab1791)<br>Chromatin from Xenopus laevis oocytes, A431 (Human epithelial carcinoma cell line) Whole Cell Lysate, Jurkat (Human T cell lymphoblast-like cell line) Whole Cell Lysate, HEK293 (Human embryonic kidney cell line) Whole Cell Lysate<br>(https://www.abcam.com/histone-h3-antibody-nuclear-marker-and-chip-grade-ab1791.html)<br><br>anti-Sox2 antibody (abcam ab 92494)<br>NCCIT (human pluripotent embryonic carcinoma cell line) whole cell lysate, MCF7 (human breast adenocarcinoma cell line) whole cell lysate, Human glioma lysate<br>(https://www.abcam.com/sox2-antibody-epr3131-ab92494.html)<br><br>HRP-conjugated anti-rabbit secondary antibody (Biorad, 170-6515):<br>https://www.bio-rad.com/en-us/sku/1662408EDU-secondary-antibody-goat-anti-rabbit-antibody-conjugated-horseradish-peroxidase?ID=1662408EDU |

# Eukaryotic cell lines

Policy information about cell lines and Sex and Gender in Research

| | |
|---|---|
| Cell line source(s) | Flp-In™ T-REx™ 293 Cell Line (Catalogue # R78007) from Thermo Fischer Scientific |
| Authentication | Authenticated by STR profiling at St Jude Children's Research Hospital |
| Mycoplasma contamination | Tested negative for Mycoplasma contamination. |
| Commonly misidentified lines<br>(See ICLAC register) | None |

# Palaeontology and Archaeology

| | |
|---|---|
| Specimen provenance | NA |
| Specimen deposition | NA |
| Dating methods | NA |

☐ Tick this box to confirm that the raw and calibrated dates are available in the paper or in Supplementary Information.

| | |
|---|---|
| Ethics oversight | NA |

Note that full information on the approval of the study protocol must also be provided in the manuscript.

# Animals and other research organisms

Policy information about studies involving animals; ARRIVE guidelines recommended for reporting animal research, and Sex and Gender in Research

| | |
|---|---|
| Laboratory animals | NA |
| Wild animals | NA |
| Reporting on sex | NA |
| Field-collected samples | NA |
| Ethics oversight | NA |

Note that full information on the approval of the study protocol must also be provided in the manuscript.

## Clinical data

Policy information about clinical studies
All manuscripts should comply with the ICMJE guidelines for publication of clinical research and a completed CONSORT checklist must be included with all submissions.

| | |
|---|---|
| Clinical trial registration | NA |
| Study protocol | NA |
| Data collection | NA |
| Outcomes | NA |

## Dual use research of concern

Policy information about dual use research of concern

### Hazards

Could the accidental, deliberate or reckless misuse of agents or technologies generated in the work, or the application of information presented in the manuscript, pose a threat to:

No | Yes
- ☒ ☐ Public health
- ☒ ☐ National security
- ☒ ☐ Crops and/or livestock
- ☒ ☐ Ecosystems
- ☒ ☐ Any other significant area

### Experiments of concern

Does the work involve any of these experiments of concern:

No | Yes
- ☒ ☐ Demonstrate how to render a vaccine ineffective
- ☒ ☐ Confer resistance to therapeutically useful antibiotics or antiviral agents
- ☒ ☐ Enhance the virulence of a pathogen or render a nonpathogen virulent
- ☒ ☐ Increase transmissibility of a pathogen
- ☒ ☐ Alter the host range of a pathogen
- ☒ ☐ Enable evasion of diagnostic/detection modalities
- ☒ ☐ Enable the weaponization of a biological agent or toxin
- ☒ ☐ Any other potentially harmful combination of experiments and agents

## ChIP-seq

### Data deposition

☐ Confirm that both raw and final processed data have been deposited in a public database such as GEO.

☐ Confirm that you have deposited or provided access to graph files (e.g. BED files) for the called peaks.

| | |
|---|---|
| Data access links<br>*May remain private before publication.* | NA |
| Files in database submission | NA |
| Genome browser session<br>(e.g. UCSC) | NA |

### Methodology

| | |
|---|---|
| Replicates | NA |

| Sequencing depth | NA |
|---|---|
| Antibodies | NA |
| Peak calling parameters | NA |
| Data quality | NA |
| Software | NA |

# Flow Cytometry

## Plots

Confirm that:

- [ ] The axis labels state the marker and fluorochrome used (e.g. CD4-FITC).
- [ ] The axis scales are clearly visible. Include numbers along axes only for bottom left plot of group (a 'group' is an analysis of identical markers).
- [ ] All plots are contour plots with outliers or pseudocolor plots.
- [ ] A numerical value for number of cells or percentage (with statistics) is provided.

## Methodology

| Sample preparation | NA |
|---|---|
| Instrument | NA |
| Software | NA |
| Cell population abundance | NA |
| Gating strategy | NA |

- [ ] Tick this box to confirm that a figure exemplifying the gating strategy is provided in the Supplementary Information.

# Magnetic resonance imaging

## Experimental design

| Design type | NA |
|---|---|
| Design specifications | NA |
| Behavioral performance measures | NA |

## Acquisition

| Imaging type(s) | NA |
|---|---|
| Field strength | NA |
| Sequence & imaging parameters | NA |
| Area of acquisition | NA |

Diffusion MRI      [ ] Used      [×] Not used

## Preprocessing

| Preprocessing software | NA |
|---|---|
| Normalization | NA |
| Normalization template | NA |
| Noise and artifact removal | NA |

| Volume censoring | NA |

## Statistical modeling & inference

| Model type and settings | NA |

| Effect(s) tested | NA |

Specify type of analysis: ☐ Whole brain ☐ ROI-based ☐ Both

| Statistic type for inference<br>(See Eklund et al. 2016) | NA |

| Correction | NA |

## Models & analysis

| n/a | Involved in the study |
|-----|----------------------|
| ☒ ☐ | Functional and/or effective connectivity |
| ☒ ☐ | Graph analysis |
| ☒ ☐ | Multivariate modeling or predictive analysis |

