## [Peer Review File · Nature]

Manuscript Title: Histone modifications regulate pioneer transcription factor cooperativity

Reviewer Comments & Author Rebuttals

Reviewer Reports on the Initial Version:

Referees' comments:

Referee #1 (Remarks to the Author):

This manuscript explores how pioneering factors OCT4 and SOX2 recognize and act on nucleosomes bound to an endogenous DNA sequence. Prior structural work was done with artificial DNA sequences that were selected for their stability and revealed direct interactions between OCT4 and the histone proteins. Here the authors show that OCT4 stabilizes a specific Lin28B phasing along the nucleosome surface using cryo-EM and biochemical experiments, and show that the primary observed OCT4 binding site is on extranucleosomal DNA. The DNA phasing enforced by OCT4 enables SOX2 binding. The authors also show that OCT4 binding is affected by H3K27ac, linking histone modification with pioneer factor binding. One issue that should be addressed is the use of the word remodel throughout the manuscript. In the nucleosome field, "remodel" typically explains behavior that involves a protein or protein complex that uses ATP to physically move DNA around histone octamers (or remove histone proteins). OCT4 does appear to be remodeling but rather appears to delimit the phasing of the histone octamer on the DNA sequence. This difference in terminology does not take away from the novelty of the paper, but should be addressed to ensure the behavior of OCT4 is properly described. This paper brings new conceptual advances to the field by showing how multiple pioneering factor binding sites can work together on a single nucleosome, and shows how this is done on a native sequence by cryo-EM, which has not been feasible in the field thus far. Specific comments that the authors should address are raised below.

Specific comments:

1-The authors state that the nucleosome alone cryo-EM sample did not have a clear sequence register for the DNA. Supplementary Figure 3A indicates that the authors have only tried to classify on the histone protein and on the entry/exit site DNA. Have the authors further classified their almost 1,000,000 canonical nucleosome particles to see if there are preferred sequence states? It may be that the state observed in the SOX2/OCT4 reconstruction is already present but to a lesser extent. For the arguments raised in this paper, it is critical for the authors to present more information on the extent of data processing done for the nucleosome alone structure. The authors should sort their particles on the DNA to potentially reveal which DNA binding states are favored in the reconstruction. Without this, it is hard to know to what extent OCT4 binding is actually "remodeling" the nucleosome structure. It is more likely that OCT4 binding stabilizes the state observed in their cryo-EM structure.

2- The authors must show some kind of experimental proof that there is actual remodeling going on as opposed to enforcement of a particular state that is sampled naturally by the nucleosome while it is on a particular DNA sequence, particularly because OCT4 was incubated with the histone proteins,

and DNA for 72hrs, during the time the nucleosome was actually formed for cryo-EM studies. If the authors cannot provide compelling remodeling data, it is suggested that the language be adapted to indicate that OCT4 potentially favors/enforces a particular conformation of the DNA on the histone octamer.

3-Figure 2A. The H4D24 interaction is stated to potentially change the compaction state of the chromatin. It is unclear what interactions are altered in the remodeled state. The authors should indicate if specific hydrogen bonds are broken and show this as a panel in the supplementary material.

4-Figure 2B/Supplementary Figures 5A, 5D: The compaction assays performed with Mg²⁺ are a bit difficult to interpret. It appears that the Δ N tail is not associating as well with the nucleosome as the WT protein (upon addition of Mg²⁺ the OCT4-nucleosome complex band immediately disappears and there is a 5x excess of OCT4 relative to the nucleosome). The provided binding measurements (native gels) appear to show some differences in binding behavior. Namely, the bands for the WT protein and Δ C OCT4 mutant are more focused than that of the OCT4 Δ Ntail. Have the authors tried a more quantitative and solution based technique like FA or ITC to measure binding affinity between the WT OCT4/the OCT4 mutants, and the nucleosome? This would better clarify if the binding affinities are indeed the same or altered in the mutants. It also appears that in the WT OCT4 experiment, the band of the OCT4-NCP complex is disappearing faster than the nucleosome alone band. These issues make the results of this experiments difficult to parse. Additionally, the authors should provide some kind of statistical test to confirm that the differences they are observing are statistically relevant (Figure 2B).

5-Supplementary Figure 4D- The resolution of the presented data is likely not the reported 4.2 Å (secondary structure should be apparent, maps are very noisy). The authors should provide an FSC curve for the data. Additional data are likely required to make the claims the authors are proposing given the limited size of the collected cryo-EM data. It is suggested that without additional data and more refined features in the cryo-EM maps, the figure and corresponding text in the manuscript be removed because they do not add substantively to the manuscript.

6-The authors refer to the N-terminal region of OCT4 that binds the nucleosome as disordered. They apparently see a fragment of it bound in their maps. It would be best to indicate the identity of this fragment if known. Additionally, the authors should refer to this region as flexible. Disordered implies that it lacks order entirely, and in the presented data the protein appears to partially gain order when bound to the nucleosome.

7-Supplementary figures: All cryo-EM reconstructions require some information about particle orientation. Currently, there is no information about this in any figure (3D FSC and/or particle distribution plot). The authors should also provide more complete cryo-EM data processing trees. These do not need to be extensive, but should give a general overview of the processing strategy employed for each sample. Representative 2D classes and micrographs should also be provided for each data set. This will allow readers to understand the differences in particle density as this is important for some of the claims made in the manuscript (see point 5).

8-Methods: The authors describe two different assembly methods to prepare samples for cryo-EM. It is unclear which samples were prepared with which method. This should be more clearly stated in the methods section.

9-Figure 3F: The authors provide a restriction enzyme digest assay to indicate the positioning of the nucleosome along the DNA in the presence of OCT4. The authors use the data in these assays to suggest that OCT4 is moving the DNA along the histone octamer surface. They have not provided data that directly indicate that OCT4 is physically sliding the nucleosome. The data more likely imply that OCT4 is stabilizing particular states rather than physically moving the DNA.

Minor issues

-Addition of Sox2/Oct4 appears to result in the formation of hexasomes (S1A, S1C). The authors should indicate in their native PAGE experiments the hexasome species that they observe in their data. It may be useful to comment on this in the manuscript. It appears that a somewhat larger population of the sample is found without the H2A/H2B dimer in the nucleosome sample that was prepared with SOX2/OCT4 in comparison to the sample prepared without.

-All figures that contain a molecular weight marker or DNA bp marker need labels for the marker. It is otherwise unclear what mass/size the bands correspond to (eg figure S1A).

-Rotation axis is needed for Supplementary Figure 1D

-Figure 2A.- Very unclear what is shown in the figure. It would help if labels were colored to match the feature they are marking (blue for canonical, black for remodeled, green label for OCT4 region).

-Methods- SYBR Gold is frequently written as SYBER Gold. The authors should correct this typo.

-Supplementary Figure 7C- mark the position of the free DNA on the right side of the gel.

-Supplementary Figure 2C- The FSC curve here looks odd. The authors should check their parameters for density modification carefully.

Referee #2 (Remarks to the Author):

In the manuscript entitled "Histone modifications regulate pioneer transcription factor binding and cooperativity", Sinha et al. study binding of OCT4 (POU5F1) and SOX2 proteins to a LIN28 nucleosome. They use mainly biochemical techniques, but also provide partial support for their model using cryo-electron microscopy. Most convincing part of the work is the cryo-EM and initial biochemistry, using which they show that one OCT4 protein first engages the nucleosome and facilitates a register shift. Using biochemical methods, they also provide evidence that two additional OCT4 proteins then bind to the same nucleosome, presumably at the sites described earlier. They also provide some biochemical evidence that one histone modification (not

modifications as claimed in the title) affect binding of the other OCT4 molecules.

The manuscript is in general well written, and the experiments proceed in a logical manner. Many findings reported are interesting, but the evidence to support them remains preliminary. The cryo-EM results are different from an earlier study that mapped optimal OCT4-SOX2 binding sites and solved their structure. That study used an in vitro selected strong nucleosomal positioning sequence. Here, the authors instead use an endogenous sequence from the LIN28 locus that is associated with (but not required for) cellular reprogramming. This makes the work of some interest. However, earlier studies have clearly shown that chromatin containing different DNA sequences will be opened with different specific mechanisms by pioneer transcription factors, making it likely that this work will not establish a general paradigm for the field. Although the specific findings are novel, they do not represent the first structure of OCT4 bound to nucleosomes, nor the first study that analyzes OCT4 binding to nucleosomes biochemically.

Due to the considerations above, and the relatively preliminary nature of the biochemical data, I feel that this work does not reach the high bar of evidence quality, novelty and generality expected of a Nature paper.

Major points

* Some evidence presented is not convincing. Association of OCT4 unstructured region with the histone tails, and its effect on nucleosome aggregation is relatively weak. More importantly, evidence highlighted in the title, the effect of histone modifications on pioneer factor cooperativity is not supported. First, there is only one histone modification. Second, the support for the effect of H3K27Ac is weak. Relatively small differences are shown between the modified and unmodified nucleosomes. EMSA assays are difficult to perform and often show high day to day variability. The authors need to show using deacetylase that the effect is specific and not due to differences in the nucleosome preparations. In addition, more replicates are needed, in which the modified and unmodified nucleosome samples are loaded to the same gel next to each other. Third, no structure or even plausible mechanism for the modification affecting the binding of the second and third OCT4 molecules are shown.

* Only LIN28 sequence is used, and the results differ from those reported by Thoma group. Without additional genomic nucleosome sequences, it is not possible to determine whether any of the results shown represent general principles of how OCT4 engages genomic nucleosomes. Based on earlier work, I would expect that picking any other OCT4-bound nucleosomal region would reveal a different particular mechanism of chromatin binding and opening.

* Many OCT4 molecules bound suggest that the protein is present in very high concentration. Failure to determine the structure of the higher order complexes also suggest that the binding is not very specific to particular positions. The authors need to establish that the concentrations of OCT4 and SOX2 used are physiologically relevant. Additional structures showing bound complexes would also help to strengthen the work.

Referee #3 (Remarks to the Author):

I read with great interest the manuscript by Sinha et al. entitled “Histone modifications regulate pioneer transcription factor binding and cooperativity”, in which the authors solve the cryoEM structure of nucleosome-bound Oct4 using a natural DNA sequence – the human Lin28B DNA - containing several Oct4 and a single Sox2 binding site. Their structural analyses show that nucleosomes by themselves are not well positioned on Lin28B DNA, but binding of Oct4 at linker-DNA position forces nucleosomes into a well-defined position. Contacts between the N-terminal trans-activation domain of Oct4 alter the position of the H4 tail with consequences for higher order chromatin structure. A second contact is detected between H3K27 and the Oct4 DBD: Interestingly acetylation of H3K27 induces a conformational change in the nucleosome, which increases binding of Oct4 and Sox2 to internal nucleosome sites due to DNA repositioning. This finding thus presents a nucleosome-mediated and histone PTM-dependent crosstalk between Oct4 and Sox2.

Together, the paper has several strong points: First, it shows a complex of a key TF with a nucleosome using natural DNA sequences, important for further progress in the chromatin field. Second, the authors show that Oct4 can ‘actively’ position – and reposition - nucleosomes on such natural DNA. Third, histone modifications can influence this behavior in a specific fashion, fine-tuning TF crosstalk. These novel findings are of great interest to the community and beyond. I thus, in principle, believe that the manuscript could be suitable for publication in Nature.

1. However, I have a question regarding the generalities of the findings: The authors claim that the positioning of H3K27ac nucleosomes is precisely altered by Oct4 interactions such that subsequent factors (Oct4, Sox, ...) exhibit improved binding (a +1 bp shift shows the effect, whereas already a +2 bp shift does not display any difference). Is this highly specific to Lin28B or can this also be observed with other sequences? And are Oct4 binding sites generally positioned at precise distances to nucleosomes (e.g. as determined by MNase seq), to have this effect? Only in this case, one would expect to observe a general correlation of Oct4 binding with H3K27ac, as mentioned in the discussion section of the paper.

2. The authors observe that the H4 tail adopts an alternative orientation on the nucleosome side close to Oct4 due to an interaction with an unidentified section of the Oct4 AD. They then relate this conformational change to changes in chromatin compaction. While I agree that the data shows that Oct4 nucleosome binding impairs chromatin compaction, and that this effect is dependent on the Oct4 N-terminus, I don't think an involvement of the H4 tail is necessary to explain this effect. Instead, Oct4 interactions with the nucleosome surface, or steric occlusion, could equally likely result in the observed effects. The authors discuss these possibilities.

- To this interpretation, Figure S4 seems important too. From this figure, it is very difficult for me to judge if the claim is valid. The authors should improve the figure, to orient the reader (e.g. include more views, so the position of Oct4 can be seen). Moreover, the authors write that the nucleosome interactions occur ‘predominantly’ on the Oct4 distal side – can this be further quantified?

3. The authors propose that H3K27ac alters the positioning of the nucleosome by 1 bp. This is indeed a very interesting, surprising and important finding. However, the authors do not directly show this

movement, but rather indirectly infer it from different experiments. There are various possible interpretation of these results, from shifts greater than 1 bp, loss of precise nucleosome positioning or structural alterations of the nucleosome due to H3K27me3.

- To sort this out, nucleosome positioning could be determined more precisely, e.g. using Dnase1 + sequencing (as in Matsumoto et al. Nature 2019), MNase-seq or via hydroxyl radical footprinting.

- It could also be informative to repeat the experiment in Figure S8c with H3K27ac octamers to see if accessibility is reduced for the Lin28B_OSO+1 sequence.

- For the quantification of the EMSAs: In general, it is not specified (or I cannot easily determine) which ratios Oct4:nucleosome (or Sox2) are quantified to generate the bar graphs, e.g. in Fig. 3B,E or 4A-C. Is it always the highest ratio? This might be hard to quantify e.g. in Fig. 3E. Conversely, there is no quantification for Fig. S8C, which seems to be critical for the interpretation of the 1 bp shift hypothesis.

4. I have issues with the analysis and interpretation of the EMSA data shown in Figure 4, in particular with the assignment of bands as well as their quantification. In short, I am completely at a loss how the authors determine that Sox2 binding is 5x stronger in Fig. 4A, or Sox2 binding in 4B 8x stronger. Moreover, it is unclear how all the bands are assigned, in particular in Figures 4B and C.

- To make the statement that Sox2 binding is strongly enhanced to Oct4-prebound nucleosomes, the authors have to come up with an alternative experimental scheme. If it must be EMSA: they could keep the Oct4 concentration constant at a level that the whole nucleosome is bound (e.g. Figure e3, ratio 4), then produce a binding curve by titrating in Sox2. Compare this to Sox2 binding to free nucleosomes. Using fluorescent markers on the TFs might aid identifying the bands.

- Are Sox2 and Oct4 directly interacting? How is this potential interaction contributing to the cooperativity, beyond the DNA positioning?

5. In general, the data lack significance testing, could easily be added.

Author Rebuttals to Initial Comments:

Referees' comments:

Referee #1 (Remarks to the Author):

This manuscript explores how pioneering factors OCT4 and SOX2 recognize and act on nucleosomes bound to an endogenous DNA sequence. Prior structural work was done with artificial DNA sequences that were selected for their stability and revealed direct interactions between OCT4 and the histone proteins. Here the authors show that OCT4 stabilizes a specific Lin28B phasing along the nucleosome surface using cryo-EM and biochemical experiments, and show that the primary observed OCT4 binding site is on extranucleosomal DNA. The DNA phasing enforced by OCT4 enables SOX2 binding. The authors also show that OCT4 binding is affected by H3K27ac, linking histone modification with pioneer factor binding. One issue that should be addressed is the use of the word remodel throughout the manuscript. In the nucleosome field, "remodel" typically explains behavior that involves a protein or protein complex that uses ATP to physically move DNA around histone octamers (or remove histone proteins). OCT4 does appear to be remodeling but rather appears to delimit the phasing of the histone octamer on the DNA sequence. This difference in terminology does not take away from the novelty of the paper, but should be addressed to ensure the behavior of OCT4 is properly described. This paper brings new conceptual advances to the field by showing how multiple pioneering factor binding sites can work together on a single nucleosome, and shows how this is done on a native sequence by cryo-EM, which has not been feasible in the field thus far. Specific comments that the authors should address are raised below.

We appreciate the reviewer's enthusiasm for our work and agree the terminology could be more precise. We have revised the manuscript to reflect that Oct4 passively stabilizes the nucleosome in a specific conformation.

Specific comments:

1-The authors state that the nucleosome alone cryo-EM sample did not have a clear sequence register for the DNA. Supplementary Figure 3A indicates that the authors have only tried to classify on the histone protein and on the entry/exit site DNA. Have the authors further classified their almost 1,000,000 canonical nucleosome particles to see if there are preferred sequence states? It may be that the state observed in the SOX2/OCT4 reconstruction is already present but to a lesser extent. For the arguments raised in this paper, it is critical for the authors to present more information on the extent of data processing done for the nucleosome alone structure. The authors should sort their particles on the DNA to potentially reveal which DNA binding states are favored in the reconstruction. Without this, it is hard to know to what extent OCT4 binding is actually "remodeling" the nucleosome structure. It is more likely that OCT4 binding stabilizes the state observed in their cryo-EM structure.

The reviewer is correct that the state captured in the Oct4-bound nucleosome structure is present in the nucleosome alone, although to a lesser extent. Thus, Oct4 indeed stabilizes a state that normally appears on the non-positioning, endogenous sequence we used. This is reflected in the revised text.

In the revision, we present more details on data processing and classification procedures. We have indeed extensively classified the data, but because the signal from DNA in different positions is small (shifts of several base pairs), the particles are difficult to separate; thus, determining the precise percentage of particles in nucleosome-alone dataset that are in the state observed in the Oct4-bound structure is very challenging.

2- The authors must show some kind of experimental proof that there is actual remodeling going on as opposed to enforcement of a particular state that is sampled naturally by the nucleosome while it is on a particular DNA sequence, particularly because OCT4 was incubated with the histone proteins, and DNA for 72hrs, during the time the nucleosome was actually formed for cryo-EM studies. If the authors

cannot provide compelling remodeling data, it is suggested that the language be adapted to indicate that OCT4 potentially favors/enforces a particular conformation of the DNA on the histone octamer.

The reviewer is correct that Oct4 binding stabilizes a state that normally appears through spontaneous nucleosome sliding and reduces further nucleosome mobility. We have revised the text accordingly.

3-Figure 2A. The H4D24 interaction is stated to potentially change the compaction state of the chromatin. It is unclear what interactions are altered in the remodeled state. The authors should indicate if specific hydrogen bonds are broken and show this as a panel in the supplementary material.

The interaction of Oct4 with H4D24 and consequent rotation of that residue lead to a different path for the H4 tail, compared to the canonical nucleosome, thus altering the positions of residues essential for interaction with another nucleosome, such as H4K16, H4R19 or H4K20. We observe that H4D24 interacts with additional density originating from Oct4, but the resolution of the map does not allow us to describe the interaction in more details.

4-Figure 2B/Supplementary Figures 5A, 5D: The compaction assays performed with Mg²⁺ are a bit difficult to interpret. It appears that the Δ N tail is not associating as well with the nucleosome as the WT protein (upon addition of Mg²⁺ the OCT4-nucleosome complex band immediately disappears and there is a 5x excess of OCT4 relative to the nucleosome). The provided binding measurements (native gels) appear to show some differences in binding behavior. Namely, the bands for the WT protein and Δ C OCT4 mutant are more focused than that of the OCT4 Δ Ntail. Have the authors tried a more quantitative and solution based technique like FA or ITC to measure binding affinity between the WT OCT4/the OCT4 mutants, and the nucleosome? This would better clarify if the binding affinities are indeed the same or altered in the mutants. It also appears that in the WT OCT4 experiment, the band of the OCT4-NCP complex is disappearing faster than the nucleosome alone band. These issues make the results of this experiments difficult to parse. Additionally, the authors should provide some kind of statistical test to confirm that the differences they are observing are statistically relevant (Figure 2B).

In the revised version, we have investigated the binding of Oct4 Δ N to nucleosomes in more details, using additional points in the dose-response native gel analysis, thus providing a more rigorous quantitative assessment. The new data show that Oct4 Δ N and Oct4 bind nucleosomes in a very similar manner (Figure ED 5d). We have also performed statistical analyses (Student's t-test) to show whether differences are statistically significant in Figure 2b and all other appropriate panels throughout the paper.

The reviewer mentions that Oct4-nucleosome band disappearing faster than nucleosome alone band (Figure ED 5a), but this small apparent variation is not observed in all assays. What we do observe consistently is that upon Mg²⁺ addition, the nucleosome band disappears faster in samples without Oct4 or with Oct4 Δ N, compared to samples with Oct4 or Oct4 Δ C (Figure ED 5a). Since we use 5x excess Oct4 in those experiments (40 nM nucleosomes; 200 nM Oct4), as the reviewer points out, we think that free nucleosomes are also transiently bound by Oct4, which reduces inter-nucleosome interactions and precipitation. Oct4 residency time on nucleosome was estimated to be in the range of ~40s (Li et al, 2019 Cell Reports), suggesting that Oct4 might transiently engage "free" nucleosomes in solution, which would affect their compaction, but it might dissociate prior to or during gel electrophoresis.

5-Supplementary Figure 4D- The resolution of the presented data is likely not the reported 4.2 Å (secondary structure should be apparent, maps are very noisy). The authors should provide an FSC curve for the data. Additional data are likely required to make the claims the authors are proposing given the limited size of the collected cryo-EM data. It is suggested that without additional data and more refined features in the cryo-EM maps, the figure and corresponding text in the manuscript be removed because they do not add substantively to the manuscript.

The structure in the original Suppl. Fig. 4D was shown at low contour level, to show the density of the flexible Oct4, therefore the secondary structure elements were not visible. We have followed the reviewer's suggestion and removed this structure and corresponding text, as we agree they do not add much.

In the revised version, we have analyzed inter-nucleosome interactions in the Oct4-bound dataset and found that Oct4 binding changes how two nucleosomes interacted (Figure ED 4d). In a subset of particles without Oct4, nucleosomes interacted predominantly near the linker DNA. In the Oct4-bound subset, on the Oct4 distal side, nucleosomes interacted predominantly at the histone octamer interface in different orientations. On the Oct4 proximal side, nucleosomes predominantly interacted at the back of the nucleosome, away from Oct4 and linker DNA. These data show that both Oct4 DNA binding domain and N-terminal tail contribute to interactions between two nucleosomes.

6-The authors refer to the N-terminal region of OCT4 that binds the nucleosome as disordered. They apparently see a fragment of it bound in their maps. It would be best to indicate the identity of this fragment if known. Additionally, the authors should refer to this region as flexible. Disordered implies that it lacks order entirely, and in the presented data the protein appears to partially gain order when bound to the nucleosome.

We agree with the reviewer and replaced the term "disordered" with "flexible". We do not know the exact residues interacting since the resolution for the density is too low.

7-Supplementary figures: All cryo-EM reconstructions require some information about particle orientation. Currently, there is no information about this in any figure (3D FSC and/or particle distribution plot). The authors should also provide more complete cryo-EM data processing trees. These do not need to be extensive, but should give a general overview of the processing strategy employed for each sample. Representative 2D classes and micrographs should also be provided for each data set. This will allow readers to understand the differences in particle density as this is important for some of the claims made in the manuscript (see point 5).

We appreciate the reviewer's suggestions and have provided more extensive information on data analyses in the revision. We also describe the extensive classification performed, in summarized form, due to space constraints.

8-Methods: The authors describe two different assembly methods to prepare samples for cryo-EM. It is unclear which samples were prepared with which method. This should be more clearly stated in the methods section.

We have clarified this matter in the revised methods section.

9-Figure 3F: The authors provide a restriction enzyme digest assay to indicate the positioning of the nucleosome along the DNA in the presence of OCT4. The authors use the data in these assays to suggest that OCT4 is moving the DNA along the histone octamer surface. They have not provided data that directly indicate that OCT4 is physically sliding the nucleosome. The data more likely imply that OCT4 is stabilizing particular states rather than physically moving the DNA.

As previously stated, the reviewer is correct: our data indicate that Oct4 does not actively move DNA, but rather stabilizes states attained by intrinsic nucleosome mobility. We have revised the text to clarify the mechanism proposed and avoid any misunderstanding.

Minor issues

-Addition of Sox2/Oct4 appears to result in the formation of hexasomes (S1A, S1C). The authors should indicate in their native PAGE experiments the hexasome species that they observe in their data. It may be useful to comment on this in the manuscript. It appears that a somewhat larger population of the sample is found without the H2A/H2B dimer in the nucleosome sample that was prepared with SOX2/OCT4 in comparison to the sample prepared without.

The reviewer is correct: hexasomes correspond to 5% of the particles in the sample with Sox2/Oct4. We mention this observation in the revision. However, under the conditions used in native PAGE experiments, it would be difficult to observe the small population of hexasomes.

-All figures that contain a molecular weight marker or DNA bp marker need labels for the marker. It is otherwise unclear what mass/size the bands correspond to (eg figure S1A).

-Rotation axis is needed for Supplementary Figure 1D

-Figure 2A.- Very unclear what is shown in the figure. It would help if labels were colored to match the feature they are marking (blue for canonical, black for remodeled, green label for OCT4 region).

-Methods- SYBR Gold is frequently written as SYBER Gold. The authors should correct this typo.

-Supplementary Figure 7C- mark the position of the free DNA on the right side of the gel.

-Supplementary Figure 2C- The FSC curve here looks odd. The authors should check their parameters for density modification carefully.

These were all done in the revision: we improved the labeling in the figures, corrected typos, and replaced the FSC curve.

Referee #2 (Remarks to the Author):

In the manuscript entitled "Histone modifications regulate pioneer transcription factor binding and cooperativity", Sinha et al. study binding of OCT4 (POU5F1) and SOX2 proteins to a LIN28 nucleosome. They use mainly biochemical techniques, but also provide partial support for their model using cryo-electron microscopy. Most convincing part of the work is the cryo-EM and initial biochemistry, using which they show that one OCT4 protein first engages the nucleosome and facilitates a register shift. Using biochemical methods, they also provide evidence that two additional OCT4 proteins then bind to the same nucleosome, presumably at the sites described earlier. They also provide some biochemical evidence that one histone modification (not modifications as claimed in the title) affect binding of the other OCT4 molecules.

The manuscript is in general well written, and the experiments proceed in a logical manner. Many findings reported are interesting, but the evidence to support them remains preliminary. The cryo-EM results are different from an earlier study that mapped optimal OCT4-SOX2 binding sites and solved their structure. That study used an in vitro selected strong nucleosomal positioning sequence. Here, the authors instead use an endogenous sequence from the LIN28 locus that is associated with (but not required for) cellular reprogramming. This makes the work of some interest. However, earlier studies have clearly shown that chromatin containing different DNA sequences will be opened with different specific mechanisms by pioneer transcription factors, making it likely that this work will not establish a general paradigm for the field. Although the specific findings are novel, they do not represent the first structure of OCT4 bound to nucleosomes, nor the first study that analyzes OCT4 binding to nucleosomes biochemically.

Due to the considerations above, and the relatively preliminary nature of the biochemical data, I feel that this work does not reach the high bar of evidence quality, novelty and generality expected of a Nature paper.

The reviewer is correct that there is a previous structure of OCT4 bound to a nucleosome (Michael et al, 2020), but that study used nucleosomes assembled with the 601 positioning sequence with binding

sites for Oct4-Sox2 inserted at a specific place, resulting in a fixed position for Oct4-Sox2.

In contrast, our structure is the first of Oct4 bound to a nucleosome with an endogenous DNA sequence, which is an important distinction in our view. The endogenous DNA sequence allows the nucleosome to move and can undergo passive “repositioning” in response to Oct4 binding; this leads to additional interactions between Oct4 and histones that would not be possible if Oct4 binds at a fixed position. Thus, our findings are entirely novel and underscore the complexity of nucleosome transactions and the need to use physiologically relevant substrates.

The reviewer questions whether our work represents a general paradigm for the field. We do not claim Oct4 binding affects all nucleosomes in the same way it affects the Lin28B nucleosome. In fact, that seems unlikely, as it would require the binding motifs for Oct4 and Sox2 to be in the same relative positions on the DNA and on the nucleosome, as well as the same dynamic behavior of that DNA sequence on the nucleosome. Instead, our work reveals how cooperativity among pioneer transcription factors can occur and provide the first evidence (to our knowledge) that a histone modifications can affect transcription factor binding, a finding that likely applies to many chromatin loci, though the specific mechanisms might vary across different DNA sequence or chromatin contexts, wherein H3K27ac or other histone modifications might increase or decrease transcription factor cooperativity.

In fact, in the revision, we present new data showing the impact of H3K27ac on Oct4 binding to nucleosomes reconstituted with another DNA sequence, identified from previous Oct4 ChIP-seq studies (Soufi et al, Cell, 2012). We note that the only endogenous DNA sequence well characterized for Oct4 binding is that of Lin28B. The new sequence examined contains multiple potential binding sites for Oct4, and we are able to detect 3-4 molecules of Oct4 bound to nMatn1 nucleosome (Figure 4c-d). Acetylation or methylation of H3K27 increases the binding of the 3rd and 4th molecules of Oct4 to nMatn1 nucleosome (Figure 4c-d). These additional binding assays demonstrate that the principle we uncovered with the Lin28B nucleosome — namely, that histone modification can affect binding of a transcription factor — applies to other DNA sequences.

In the revision we also present a new cryo-EM structure of Oct bound to nMatn1 nucleosome (Figure 4a-b, ED 8 and 9). Oct4 binds nMatn1 nucleosome at a location similar to that on the Lin28B nucleosome and interacts with the H3 tail through the acidic patch of Oct4 S domain. The structure reveals similarities, but also differences to Lin28B structure, which are described in detail in the revised manuscript.

Major points

* Some evidence presented is not convincing. Association of OCT4 unstructured region with the histone tails, and its effect on nucleosome aggregation is relatively weak. More importantly, evidence highlighted in the title, the effect of histone modifications on pioneer factor cooperativity is not supported. First, there is only one histone modification. Second, the support for the effect of H3K27Ac is weak. Relatively small differences are shown between the modified and unmodified nucleosomes.

To strengthen our finding that Oct4 binding reduces nucleosome interactions, in the revision we provide additional data characterizing Oct4 mutants with deletions in the unstructured region (Figure ED 5d), as well as statistical analysis of the biochemistry data. We also provide additional analysis of cryo-EM data to show that inter-nucleosome interactions are different when Oct4 is bound (Figure ED 4d): Oct4 binding reduces interactions near the linker DNA and at the histone octamer surface.

We disagree that the data supporting the effect of H3K27ac is weak: on average, we observe ~6 fold increase in binding of 3rd Oct4 and ~6-fold in Sox2 binding, which can have huge effects on transcription activation. These effects are robust and we have performed the appropriate statistical analyses.

To investigate further the effect of histone modification on TF binding, in the revision, we also

assessed Oct4 binding to nucleosomes containing another modification, H3K27me3. Our data show that effect of H3K27me3 is different to that of H3K27ac, with only a minor effect on Oct4 interactions with the Lin28B nucleosome (Figure ED 7a). In contrast, H3K27me3 increased binding of Oct4 to internal sites on the nMatn1 nucleosome (Figure 4d). These data show that different histone modifications can result in different outcomes on transcription factor binding and cooperativity.

EMSA assays are difficult to perform and often show high day to day variability. The authors need to show using deacetylase that the effect is specific and not due to differences in the nucleosome preparations. In addition, more replicates are needed, in which the modified and unmodified nucleosome samples are loaded to the same gel next to each other. Third, no structure or even plausible mechanism for the modification affecting the binding of the second and third OCT4 molecules are shown.

It is indeed our experience that EMSA can show day-to-day variation. To overcome that caveat, we have followed the reviewer's suggestions: we performed all related assays at the same time and loaded the samples on the same gel. In addition, to minimize potential effects due to different nucleosome preparations, we increased the number of replicates and quantified results from 3-6 independent experiments.

As requested by the reviewer, to rule out effects from different nucleosome preparations, we deacetylated H3K27ac nucleosomes using MiDAC and used those nucleosomes in our EMSA assays. We observed that second and third Oct4 bind better H3K27ac nucleosome when compared to deacetylated H3K27ac nucleosome (Figure ED 6b, c), similar to unmodified nucleosomes.

Regarding the mechanism by which H3K27ac affects cooperative Oct4 binding, the structural findings provide us with such explanation. The unmodified, positively charged H3K27 interacts with the acidic patch of Oct4 bound at OBS1 (Figure 3d). Acetylation of H3K27 removes the positive charge and prevents the electrostatic interaction between H3K27 and Oct4, leading to changes in DNA positioning on the nucleosome and different exposure of internal binding sites.

These structural findings are supported by our biochemistry analyses. The Mnl I restriction data (Figure 3f) show higher protection of Mnl I site when Oct4 is bound to H3K27ac nucleosome. In the revision, we have also sequenced MNase-digested DNA from Oct4-bound nucleosomes that were unmodified or containing H3K27ac (Figure 3g, ED 6f). Our data show that H3K27ac leads to DNA re-positioning by 1-2 bp, which better exposes internal binding sites, especially the Sox2 binding site.

* Only LIN28 sequence is used, and the results differ from those reported by Thoma group. Without additional genomic nucleosome sequences, it is not possible to determine whether any of the results shown represent general principles of how OCT4 engages genomic nucleosomes. Based on earlier work, I would expect that picking any other OCT4-bound nucleosomal region would reveal a different particular mechanism of chromatin binding and opening.

Indeed, our findings differ from those by the Thoma group, and this is an important distinction. We used a native human sequence that is targeted by Oct4 in cells, whereas the Thoma group used an artificial construct. We thus believe that our work better represents the cellular activity of Oct4.

Regarding the reviewer's concern about general principles of Oct4 engagement with genomic nucleosomes, we do not claim our findings apply to all Oct4 binding sites, as explained in our response to this reviewer's opening comments. In the revised version, we show the impact of H3K27ac and H3K27me3 on Oct4 binding to nucleosomes reconstituted with another DNA sequence, identified from previous ChIP-seq studies (Figure 4, ED 8, 9). These new binding assays demonstrate that the general principle we uncovered for the Lin28B nucleosome — namely, that histone modification can affect binding of a transcription factor — applies to other DNA sequences.

In addition, we have determined a new cryo-EM structure showing Oct4 bound to the Mat1n nucleosome, which reveals that Oct4 binds in a similar location and interacts with the H3 tail as observed for Oct4 bound to Lin28B nucleosome (Figure 4a-b, ED 8, 9). These new data strengthen

our conclusions.

* Many OCT4 molecules bound suggest that the protein is present in very high concentration. Failure to determine the structure of the higher order complexes also suggest that the binding is not very specific to particular positions. The authors need to establish that the concentrations of OCT4 and SOX2 used are physiologically relevant. Additional structures showing bound complexes would also help to strengthen the work.

For our structural work, we used low concentrations (1 μ M) of full-length Oct4, to reduce sample aggregation on the grids, with molar ratio of 1:2 to nucleosomes, and observed that ca. 40% of the nucleosomes had Oct4 bound at OBS1. Thus, these are not saturated conditions, and we did not expect to see more than 1 Oct4 bound per nucleosome. For biochemical assays, Oct4 and Sox2 concentrations were even lower, in the range of 50–400 nM. In previous work by Chen et al (2014, Cell), the authors have estimated that Sox2 concentration in cells is 730 nM, indicating that the concentrations we used in our structural and biochemical work are physiologically relevant.

Binding of Oct4 to internal sites is less stable since only one Oct4 domain (either S or HD domain) can engage with DNA, which could lead to complex dissociation during sample preparation for cryo-EM. We do aim to determine structures showing Oct4 binding to internal sites, but that is a long-term project that is out of scope for the current work.

Referee #3 (Remarks to the Author):

I read with great interest the manuscript by Sinha et al. entitled “Histone modifications regulate pioneer transcription factor binding and cooperativity”, in which the authors solve the cryoEM structure of nucleosome-bound Oct4 using a natural DNA sequence – the human Lin28B DNA - containing several Oct4 and a single Sox2 binding site. Their structural analyses show that nucleosomes by themselves are not well positioned on Lin28B DNA, but binding of Oct4 at linker-DNA position forces nucleosomes into a well-defined position. Contacts between the N-terminal trans-activation domain of Oct4 alter the position of the H4 tail with consequences for higher order chromatin structure. A second contact is detected between H3K27 and the Oct4 DBD: Interestingly acetylation of H3K27 induces a conformational change in the nucleosome, which increases binding of Oct4 and Sox2 to internal nucleosome sites due to DNA repositioning. This finding thus presents a nucleosome-mediated and histone PTM-dependent crosstalk between Oct4 and Sox2.

Together, the paper has several strong points: First, it shows a complex of a key TF with a nucleosome using natural DNA sequences, important for further progress in the chromatin field. Second, the authors show that Oct4 can ‘actively’ position – and reposition - nucleosomes on such natural DNA. Third, histone modifications can influence this behavior in a specific fashion, fine-tuning TF crosstalk. These novel findings are of great interest to the community and beyond. I thus, in principle, believe that the manuscript could be suitable for publication in Nature.

We thank the reviewer for the support.

1. However, I have a question regarding the generalities of the findings: The authors claim that the positioning of H3K27ac nucleosomes is precisely altered by Oct4 interactions such that subsequent factors (Oct4, Sox, ...) exhibit improved binding (a +1 bp shift shows the effect, whereas already a +2 bp shift does not display any difference). Is this highly specific to Lin28B or can this also be observed with other sequences? And are Oct4 binding sites generally positioned at precise distances to nucleosomes (e.g. as determined by MNase seq), to have this effect? Only in this case, one would

expect to observe a general correlation of Oct4 binding with H3K27ac, as mentioned in the discussion section of the paper.

We acknowledge the reviewer's point about the generalities of the findings. Indeed, we do not expect that Oct4 binding will affect all nucleosomes in the same way as the Lin28B nucleosome. In fact, that seems unlikely, as it would require the binding motifs for Oct4 and Sox2 to be in the same relative positions on the DNA and on the nucleosome, as well as the same dynamic behavior of that DNA sequence on the nucleosome. Nonetheless, our work reveals how cooperativity among pioneer transcription factors can occur; it also provides the first evidence (to our knowledge) that a histone modification can affect transcription factor binding, and this finding is likely to apply to many chromatin loci, though the specific effects and mechanisms might vary across different DNA sequences or chromatin contexts.

In the revision, we present new data examining the impact of H3K27ac on Oct4 binding to nucleosomes reconstituted with another DNA sequence, identified from previous ChIP-seq studies (Figure 4c, d). These data demonstrate that the general principle we uncovered for the Lin28B nucleosome (i.e., that histone modification can affect binding of a transcription factor) applies to other DNA sequences. In addition, we present a new cryo-EM structure showing Oct4 bound to nMatn1 nucleosome (Figure 4a-b, ED 8, 9), which reveals that Oct4 binds in similar position as in the Lin28b nucleosome and interacts with the H3 tail.

Regarding the +1bp shift due to H3K27ac, this was proposed as a possible explanation for our biochemistry data, as a model to show that even small shifts in DNA positioning can have a large impact on transcription factor binding. In the revised version, we present data that supports that model: we sequenced MNase-digested DNA from Oct4-bound nucleosomes and observed that Oct4 binding to H3K27ac nucleosomes indeed moves DNA inwards, compared to non-modified nucleosome, with +1 bp being the predominant population (Figure 3g, ED 6f).

2. The authors observe that the H4 tail adopts an alternative orientation on the nucleosome side close to Oct4 due to an interaction with an unidentified section of the Oct4 AD. They then relate this conformational change to changes in chromatin compaction. While I agree that the data shows that Oct4 nucleosome binding impairs chromatin compaction, and that this effect is dependent on the Oct4 N-terminus, I don't think an involvement of the H4 tail is necessary to explain this effect. Instead, Oct4 interactions with the nucleosome surface, or steric occlusion, could equally likely result in the observed effects. The authors discuss these possibilities.

We agree and discuss these possibilities in the revision.

- To this interpretation, Figure S4 seems important too. From this figure, it is very difficult for me to judge if the claim is valid. The authors should improve the figure, to orient the reader (e.g. include more views, so the position of Oct4 can be seen). Moreover, the authors write that the nucleosome interactions occur 'predominantly' on the Oct4 distal side – can this be further quantified?

We have analyzed cryo-EM data of Oct4 bound nucleosome in more details and provide new figures showing interaction between two nucleosomes and how they change after Oct4 binding (Figure ED 4d, with number of particles shown). While free nucleosomes interact (9% of all particles) predominantly near the linker DNA, this interaction mode is undetectable in the Oct4-bound nucleosomes, indicating that Oct4 binding to DNA sterically interferes with those interactions. In the Oct4-bound particles, on the Oct4 distal side, nucleosomes interact (2.7% of nucleosomes interact) predominantly at the histone octamer interface, in different orientations. On the Oct4 proximal side, 1.5% nucleosomes interact, predominantly at the back of the nucleosome, away from Oct4 and linker DNA.

3. The authors propose that H3K27ac alters the positioning of the nucleosome by 1 bp. This is indeed a very interesting, surprising and important finding. However, the authors do not directly show this movement, but rather indirectly infer it from different experiments. There are various possible interpretation of these results, from shifts greater than 1 bp, loss of precise nucleosome positioning or structural alterations of the nucleosome due to H3K27me3.

- To sort this out, nucleosome positioning could be determined more precisely, e.g. using DnaseI + sequencing (as in Matsumoto et al. Nature 2019), MNase-seq or via hydroxyl radical footprinting.

As previously stated in response to this reviewer's point 1, the +1bp shift was proposed as a possible explanation for the biochemistry data. In the revision, we provide sequencing data for MNase-digested Oct4-bound nucleosomes, as suggested by the reviewer. Our data show that Oct4-bound H3K27ac nucleosomes are less well positioned than Oct4 bound unmodified nucleosomes, with +1 shift being the predominant state (Figure 3g).

- It could also be informative to repeat the experiment in Figure S8c with H3K27ac octamers to see if accessibility is reduced for the Lin28B_OSO+1 sequence.

We have performed this experiment and indeed see reduced Oct4 binding to H3K27ac Lin28B_OSO+1 nucleosomes (Figure ED 6j).

- For the quantification of the EMSAs: In general, it is not specified (or I cannot easily determine) which ratios Oct4:nucleosome (or Sox2) are quantified to generate the bar graphs, e.g. in Fig. 3B,E or 4A-C. Is it always the highest ratio? This might be hard to quantify e.g. in Fig. 3E. Conversely, there is no quantification for Fig. S8C, which seems to be critical for the interpretation of the 1 bp shift hypothesis.

We have added this information to the figure legend and labeled lanes that were used for quantification. We also added quantification for S8C (now ED 6h).

4. I have issues with the analysis and interpretation of the EMSA data shown in Figure 4, in particular with the assignment of bands as well as their quantification. In short, I am completely at a loss how the authors determine that Sox2 binding is 5x stronger in Fig. 4A, or Sox2 binding in 4B 8x stronger. Moreover, it is unclear how all the bands are assigned, in particular in Figures 4B and C.

*We used western blots to assign bands (Figure ED 5f, g and 7f). We explained band quantification and assignment in more details in the revised figure legends and method section. Bands that were used for quantification are marked with *.*

- To make the statement that Sox2 binding is strongly enhanced to Oct4-prebound nucleosomes, the authors have to come up with an alternative experimental scheme. If it must be EMSA: they could keep the Oct4 concentration constant at a level that the whole nucleosome is bound (e.g. Figure e3, ratio 4), then produce a binding curve by titrating in Sox2. Compare this to Sox2 binding to free nucleosomes. Using fluorescent markers on the TFs might aid identifying the bands.

We appreciate the reviewer's concern and have toned down the statement in the revision. We tried the Sox2 titration in our EMSA setup as suggested, but it was challenging to assign and quantify Sox2 under those conditions. When all nucleosomes are bound by Oct4 we observe binding of multiple Oct4, and Sox2 binds to all those different species which is complicating EMSA data analysis (Figure 3b, e). We also tried labeling Sox2, which has only one Cys at position 265 with Alexa 546, as suggested by the reviewer, but that labeling reduced Sox2 binding to nucleosomes.

- Are Sox2 and Oct4 directly interacting? How is this potential interaction contributing to the cooperativity, beyond the DNA positioning?

OBS1 (where the first Oct4 binds) and the Sox2 binding site are too distant to allow direct interaction between the transcription factors, indicating that cooperativity occurs through nucleosome positioning. However, Oct4 and Sox2 could interact at OBS3, where their binding sites are close.

5. In general, the data lack significance testing, could easily be added.

We have now performed statistical testing for data quantification, these are stated in the revised figure legend.

Reviewer Reports on the First Revision:

Referees' comments:

Referee #1 (Remarks to the Author):

The authors have addressed most of my previous concerns. Overall, the authors have added a substantial amount of data to the revision that helps to better bolster their arguments. It is also great that they were able to include data for the nMatn1 DNA sequence and show that OCT4 appears to behave similarly on this sequence in regards to PTMs. I have a few comments that the authors should address prior to publication. These issues are relatively minor.

- Extended data figure 1b-c, 3a, b, 8b-c, — add a scale bar to the images.
- Extended data figure 8h- this is not 5.7Å resolution. The DNA should look like DNA (helical but no apparent bases visible). Similarly, OCT4 should have some kind of secondary structure.
- Extended data figure 8k- it would help if the protein depicted as well as the range of residues was marked on the structure (same for DNA bases)
- Extended data figure 4d- This figure looks artifactual. I would leave this data out given the other data in the paper that supports the idea that OCT4 loosens nucleosomes. (Also additional text that was added pg. 2) Cryo-EM is prone to producing aggregated protein due to the freezing conditions and it is hard to compare aggregation between grids. For this analysis to be done comprehensively, the various samples would have to be imaged on different grids and averaged (3-5 independent grids per sample).
- Classification trees for each structure are needed (number of particles, resolution, what was used for subsequent focused classification/refinement).

Additional comments for issues raised by reviewer #3:

Point 2, part 2- The authors use figure ED4D to justify the claim that OCT4 may alter interactions between nucleosomes. The figure is not very convincing. It is also very difficult to quantify this kind of data. I would recommend leaving this out of the text as it is a point that is not easily validated and does not change the main findings of the manuscript.

Point 3, part 2- here the data are not necessarily supportive of the claim that there is reduced binding of OCT4 to the H3K27ac Lin28B_OS0 nucleosome (no noticeable difference in binding).

Point 4- The authors use an unusual way to quantify their gel based images. The bands are not being compared against a common denominator rather they are being compared against the other species. This makes it very difficult to understand the relative binding affinities/occupancies for each state. It would help if the authors provided an additional analysis where they quantified each species relative to the the starting nucleosome intensity. They should also provide information on the box size and what value was to measure band intensity (also need to mention the software in the text).

Point 4, second part- the authors attempted to address this issue but were not able to do so using their experimental scheme. They have toned down their language about the cooperatively between

SOX2 and OCT4, so this should be adequate to meet the reviewer concern.

Point 4, third part- the authors have not really addressed the point of a direct interaction between OCT4 and SOX2. Minimally, they could show in the absence of DNA/nucleosome, that the proteins directly interact under their experimental conditions (one experiment is needed eg., pull-down, size exclusion, SPR, BLI). The authors have suggested which positions on their sequence could support a direct interaction (OBS3) and largely suggest that the cooperativity they observe is due to nucleosome positioning rather than direct protein-protein interactions.

Referee #2 (Remarks to the Author):

Sinha et al. have revised their manuscript, and addressed some of my initial criticisms by providing additional structural data that indicates that, as expected from prior biochemical studies, Oct4 utilizes different particular mechanisms for binding to different genomic sequences. The manuscript has improved with regard to the mechanistic understanding of Oct4 nucleosome binding (by stabilizing existing DNA positioning). Despite the added data, I still feel that the manuscript does not provide sufficient evidence to justify its title, and is therefore not appropriate for publication.

There are two major concerns:

The effects shown are small in magnitude, not backed up by mechanistic data, and of questionable physiological relevance. The modification studied H3K27Ac does not have a major impact in vivo on ES cells (Sankar et al. Nature Genetics 54:754-760, 2022). The sequences used are derived from the genome, but their physiological relevance is not clear (even the genes they are linked to are not required for reprogramming). Any physiological role for the motifs bound could easily be tested using CRISPR/CAS by mutating the motifs, but given the evidence shown in Sankar et al., I would not hold out much hope that the effects shown by the authors, if they occur in vivo, would have physiological impact.

The writing of the manuscript is not balanced, and the interpretation of the results goes way beyond what is actually shown. For example, there is no evidence for example for the statement in the abstract (and discussion) that "epigenetic landscape can regulate Oct4 activity to ensure proper cell reprogramming". In addition, the authors' discussion of relative merits of using synthetic and genomic sequences is very unbalanced. As noted above, they exaggerate the importance of the genomic sequence used. Furthermore, the use of genomic sequence does not ensure that the findings are physiologically relevant, as the conditions used are not physiological. Synthetic design such as that used by the Thoma group cannot be characterize as simply inferior as it identifies optimal sequences and mechanisms. Such assays address a different question that can be physiologically either less or more relevant than evidence obtained using particular genomic sequences.

Author Rebuttals to First Revision:

Point-by-point response

Referee #1 (Remarks to the Author):

The authors have addressed most of my previous concerns. Overall, the authors have added a substantial amount of data to the revision that helps to better bolster their arguments. It is also great that they were able to include data for the nMatn1 DNA sequence and show that OCT4 appears to behave similarly on this sequence in regards to PTMs. I have a few comments that the authors should address prior to publication. These issues are relatively minor.

We appreciate the reviewer's support.

*-Extended data figure 1b-c, 3a, b, 8b-c, — add a scale bar to the images.
Scale bars have been added.*

-Extended data figure 8h- this is not 5.7Å resolution. The DNA should look like DNA (helical but no apparent bases visible). Similarly, OCT4 should have some kind of secondary structure.

The reviewer is correct, that resolution is overestimated. Meanwhile we improved the density, which has now clearer appearance of DNA helix and protein better shows tubular like densities which resemble alpha helices. Estimated resolution is ~8Å. We agree that those densities are not perfect, but we need to point out that we refined something of ~20kDa in size which is very small for cryo-EM and we find that resolving secondary structure elements is already huge achievement. Most importantly, the map is of sufficient quality to dock and assign Oct4 domains.

*-Extended data figure 8k- it would help if the protein depicted as well as the range of residues was marked on the structure (same for DNA bases)
This was done.*

-Extended data figure 4d- This figure looks artifactual. I would leave this data out given the other data in the paper that supports the idea that OCT4 loosens nucleosomes. (Also additional text that was added pg. 2) Cryo-EM is prone to producing aggregated protein due to the freezing conditions and it is hard to compare aggregation between grids. For this analysis to be done comprehensively, the various samples would have to be imaged on different grids and averaged (3-5 independent grids per sample).

We appreciate the reviewer's concern and have followed their advice to remove Extended Data Figure 4d and associated text.

-Classification trees for each structure are needed (number of particles, resolution, what was used for subsequent focused classification/refinement).

This was done and additional trees are presented in ED Figure 2a-b, 3g, 8g, 9a.

Additional comments for issues raised by reviewer #3:

Point 2, part 2- The authors use figure ED4D to justify the claim that OCT4 may alter interactions between nucleosomes. The figure is not very convincing. It is also very difficult to quantify this kind of data. I would recommend leaving this out of the text as it is a point that is not easily validated and does not change the main findings of the manuscript.

This is the same point as above, and we have removed Extended Data Figure 4d and associated text.

Point 3, part 2- here the data are not necessarily supportive of the claim that there is reduced binding of OCT4 to the H3K27ac Lin28B_OSO nucleosome (no noticeable difference in binding).

This is a misunderstanding, and we apologize for the lack of clarity in the response.

Oct4 binds more readily to Lin28B_OSO+1 than to Lin28B (ED Fig 6h, previously Fig S8Cc), and reviewer 3 had asked us to repeat that experiment with H3K27ac, to see if "accessibility is reduced for the Lin28B_OSO+1 sequence". The requested experiment is in ED Fig 6j, which shows Oct4 binds to H3K27ac Lin28B_OSO+1 or Lin28B in a similar manner; thus, acetylation of H3K27 abrogated the stronger binding of Oct4 to Lin28B_OSO+1 (relative to Lin28B). In our previous response, we used the same phrasing as reviewer 3 ("reduced Oct4 binding to H3K27ac Lin28B_OSO+1 nucleosomes") but we acknowledge that the phrasing was a shortcut and potentially confusing way to describe the results. We have revised it "acetylation of H3K27 abrogated the increased binding of Oct4 to Lin28B_OSO+1 relative to Lin28B nucleosome."

Point 4- The authors use an unusual way to quantify their gel based images. The bands are not being compared against a common denominator rather they are being compared against the other species. This makes it very difficult to understand the relative binding affinities/occupancies for each state. It would help if the authors provided an additional analysis where they quantified each species relative to the starting nucleosome intensity. They should also provide information on the box size and what value was to measure band intensity (also need to mention the software in the text).

We appreciate the comment and would like to clarify that in all EMSAs presented, the bands were normalized to the nucleosome input. In almost all experiments, the same bands are compared across different conditions (for example 1st Oct4 in unmodified nucleosome compared to 1st Oct4 in H3K27ac nucleosome).

The exceptions are in the gels shown in Fig 3b and ED Fig 7g, where we assess binding events where product and substrate species differ. Following the reviewer's suggestion, for those experiments, we have included the quantification of those bands relative to the starting nucleosome intensity (Supplementary Table x or Source Data Table X). In addition, we provide information on the box size and software tool used (Methods).

Point 4, second part- the authors attempted to address this issue but were not able to do so using their experimental scheme. They have toned down their language about

the cooperatively between SOX2 and OCT4, so this should be adequate to meet the reviewer concern.

We thank the reviewer.

Point 4, third part- the authors have not really addressed the point of a direct interaction between OCT4 and SOX2. Minimally, they could show in the absence of DNA/nucleosome, that the proteins directly interact under their experimental conditions (one experiment is needed eg., pull-down, size exclusion, SPR, BLI). The authors have suggested which positions on their sequence could support a direct interaction (OBS3) and largely suggest that the cooperativity they observe is due to nucleosome positioning rather than direct protein-protein interactions.

We had interpreted reviewer 3's comment in a different manner, i.e., are Sox2 and Oct4 directly interacting in the context of the nucleosome?

Regarding a direct interaction of Sox2 and Oct4 proteins in the absence of DNA or nucleosome, this has been examined before but no direct interaction could be detected by Lam et al Biochem J (2012) 448(1): 21-33. Following the reviewer's request, we also performed EMSA and co-IP experiments and were unable to detect a direct interaction between Sox2 and Oct4 (400 nM) at the concentrations used for EMSA (50-400 nM).

Referee #2 (Remarks to the Author):

Sinha et al. have revised their manuscript, and addressed some of my initial criticisms by providing additional structural data that indicates that, as expected from prior biochemical studies, Oct4 utilizes different particular mechanisms for binding to different genomic sequences. The manuscript has improved with regard to the mechanistic understanding of Oct4 nucleosome binding (by stabilizing existing DNA positioning). Despite the added data, I still feel that the manuscript does not provide sufficient evidence to justify its title, and is therefore not appropriate for publication.

There are two major concerns:

The effects shown are small in magnitude, not backed up by mechanistic data, and of questionable physiological relevance. The modification studied H3K27Ac does not have a major impact in vivo on ES cells (Sankar et al. Nature Genetics 54:754-760, 2022). The sequences used are derived from the genome, but their physiological relevance is not clear (even the genes they are linked to are not required for reprogramming). Any physiological role for the motifs bound could easily be tested using CRISPR/CAS by mutating the motifs, but given the evidence shown in Sankar et al., I would not hold out much hope that the effects shown by the authors, if they occur in vivo, would have physiological impact.

We appreciate the reviewer's concern and have carefully revised the text to avoid any overstatements regarding physiological role.

The writing of the manuscript is not balanced, and the interpretation of the results goes way beyond what is actually shown. For example, there is no evidence for example for the statement in the abstract (and discussion) that "epigenetic landscape can regulate Oct4 activity to ensure proper cell reprogramming". In addition, the authors' discussion of relative merits of using synthetic and genomic sequences is very unbalanced. As noted above, they exaggerate the importance of the genomic sequence used. Furthermore, the use of genomic sequence does not ensure that the findings are physiologically relevant, as the conditions used are not physiological. Synthetic design such as that used by the Thoma group cannot be characterized as simply inferior as it identifies optimal sequences and mechanisms. Such assays address a different question that can be physiologically either less or more relevant than evidence obtained using particular genomic sequences.

We did not mean to dismiss or minimize in any way the value of the work by the Thoma group using a 601-based sequence, which provides important insights. In our response, we aimed to point out the differences with our own work, to address the reviewer's concerns about lack of advance. Our findings with endogenous sequences provide new insights that could not have been anticipated from the Thoma work with 601. We have carefully revised the text to avoid any overstatements.

Reviewer Reports on the Second Revision:

Referees' comments:

Referee #1 (Remarks to the Author):

The authors have addressed all of my concerns.